# A reaction-diffusion network model predicts a dual role of Cactus/IκB to regulate Dorsal/NFκB nuclear translocation in *Drosophila*

Claudio D. T. Barros[1], Maira A. Cardoso[2,3], Paulo M. Bisch[2], Helena M. Araujo[3]*, Francisco J. P. Lopes[4]*

1 Laboratório Nacional de Computação Científica (LNCC), Petrópolis, Brasil, 2 Laboratório de Física Biológica, Instituto de Biofísica Carlos Chagas Filho (IBCCF), Universidade Federal do Rio de Janeiro (UFRJ), Rio de Janeiro, Brasil, 3 Laboratório de Biologia Molecular do Desenvolvimento, Instituto de Ciências Biomédicas, Universidade Federal do Rio de Janeiro (UFRJ), Centro de Ciências da Saúde, Rio de Janeiro, Brasil, 4 Grupo de Biologia do Desenvolvimento e Sistemas Dinâmicos, Campus Duque de Caxias Professor Geraldo Cidade, Universidade Federal do Rio de Janeiro (UFRJ)–Duque de Caxias, Brasil

* haraujo@histo.ufrj.br (HMA); flopes@ufrj.br (FJPL)

**Data Availability Statement:** All relevant data are within the manuscript and its Supporting Information files.

## Abstract

Dorsal-ventral patterning of the *Drosophila* embryo depends on the NFκB superfamily transcription factor Dorsal (Dl). Toll receptor activation signals for degradation of the IκB inhibitor Cactus (Cact), leading to a ventral-to-dorsal nuclear Dl gradient. Cact is critical for Dl nuclear import, as it binds to and prevents Dl from entering the nuclei. Quantitative analysis of *cact* mutants revealed an additional Cact function to promote Dl nuclear translocation in ventral regions of the embryo. To investigate this dual Cact role, we developed a predictive model based on a reaction-diffusion regulatory network. This network distinguishes non-uniform Toll-dependent Dl nuclear import and Cact degradation, from the Toll-independent processes of Cact degradation and reversible nuclear-cytoplasmic Dl flow. In addition, it incorporates translational control of Cact levels by Dl. Our model successfully reproduces wild-type data and emulates the Dl nuclear gradient in mutant *dl* and *cact* allelic combinations. Our results indicate that the dual role of Cact depends on the dynamics of Dl-Cact trimers along the dorsal-ventral axis: In the absence of Toll activation, free Dl-Cact trimers retain Dl in the cytoplasm, limiting the flow of Dl into the nucleus; in ventral-lateral regions, Dl-Cact trimers are recruited by Toll activation into predominant signaling complexes and promote Dl nuclear translocation. Simulations suggest that the balance between Toll-dependent and Toll-independent processes are key to this dynamics and reproduce the full assortment of Cact effects. Considering the high evolutionary conservation of these pathways, our analysis should contribute to understanding NFκB/c-Rel activation in other contexts such as in the vertebrate immune system and disease.

## Author summary

In *Drosophila*, Toll pathway establishes spatially distinct gene expression territories that define the embryonic dorsal-ventral axis. Toll activation leads to degradation of the IκB

**Funding:** PB, FL and HA were supported by FAPERJ - Fundação de Amparo à Pesquisa do Estado do Rio de Janeiro (www.faperj.br) [Grant no. E-26 010.001877/2015]. CB and MC were supported by PhD and Pos-Doc fellowships form CAPES and CNPq respectively. The funders had no role in study design, data collection and analysis, decision to publish, or preparation of the manuscript.

**Competing interests:** The authors have declared that no competing interests exist.

inhibitor Cactus, releasing the NFκB superfamily transcription factor Dorsal for nuclear entry. Recently, quantitative analysis of *cact* mutants revealed that Cact displays an additional function to promote Dl nuclear translocation in ventral regions of the embryo. To understand this novel activity, we developed a predictive theoretical model that shows that the kinetics of Dorsal-Cactus complex formation prior to their recruitment to Toll-signaling complexes is an essential regulatory hub. Cactus controls the balance between the recruitment of these complexes by active Toll receptor and association-dissociation events that generate free Dorsal for direct nuclear import.

## Introduction

In a developing organism, tissues are often patterned by long-range and spatially graded signaling factors called morphogens, which carry the positional information necessary to control gene expression. Morphogens act in a concentration-dependent manner to activate or repress target genes [1–4]. Therefore, precisely defining the amount of activated morphogen is crucial to determining their effects. In the *Drosophila* syncytial blastoderm, dorsal-ventral (DV) patterning depends on the nuclear localization gradient of Dorsal (Dl), an NFκB superfamily transcription factor homologous to mammalian c-Rel. Dl acts in a concentration-dependent manner to activate or repress target genes, defining three main territories of the embryo DV axis: the ventral mesoderm, lateral neuroectoderm, and dorsal ectoderm [5]. A ventral-to-dorsal activity gradient of the Toll cell surface receptor provides the activating signal for DV patterning. Dl nuclear translocation is controlled by Cactus (Cact), a cytoplasmic protein related to mammalian IκB. In the absence of Toll signals, Cact binds to Dl and impairs its nuclear translocation [6]. Activated Toll receptors on the ventral and lateral regions of the embryo lead to Cact phosphorylation followed by ubiquitination and degradation, resulting in dissociation of the Dl-Cact complex and Dl nuclear import. Different Dl levels subdivide the embryonic DV axis into target gene expression domains, defined by distinct thresholds of sensitivity to control by Dl [7]. Thus, the amount of Dl in the nuclei is key to defining ventral, lateral and dorsal territories of the embryo.

In attempts to explain the complex Dl regulatory signaling network, mathematical models were developed to simulate the Dl nuclear gradient. Early models simulated gradient profiles throughout nuclear division cycles 10 to 14 in wild-type embryos, with parameters constrained by experimental data from endogenous nuclear Dl (nDl) levels in fixed embryos or live imaging of Dl-GFP [8,9]. Their results showed that the nuclear Dl gradient is dynamic, increasing in amplitude from cycle 10 to cycle 14, without significantly changing its shape. Subsequently, it was shown that both the nDl gradient amplitude and basal levels oscillate throughout early embryonic development [10]. Recently, it was proposed that facilitated diffusion along the DV axis, or "shuttling" via Dl-Cact complexes, plays a role in nDl gradient formation [11]. Therefore, these analyses suggest that the establishment of the nDl gradient is a complex process yet to be fully understood.

Former initiatives modeling the nDl gradient did not clearly distinguish the processes regulated by Toll and those that do not depend on Toll activity. However, several proteins that impact Dl nuclear localization act independently of Toll-receptor activation. For instance, adaptor proteins, which are required to transduce Toll signals, such as Tube, Pelle and Myd88, assemble membrane-associated pre-signaling complexes in the absence of Toll [12–16]. In addition, calcium-dependent Calpain A protease activity alters Cact function and perturbs the nDl gradient [3,17]. Dl itself can flow into and out of the nucleus in the absence of Toll

activation [11,18], with the potential to contribute significantly to nDl concentration. Since Cact plays a critical role in the control of Dl nuclear translocation, understanding how Cact is regulated is paramount.

Two pathways have been reported to regulate Cact levels: the Toll dependent pathway that leads to Cact N-terminal phosphorylation and degradation through the proteasome, and a Toll-independent pathway that targets Cact C-terminal sequences for phosphorylation [19]. This originally termed "signal independent pathway for Cact degradation" [6,19–21] acts in parallel to Toll-induced signaling [22]. However, several reports indicate that the signal-independent pathway is regulated by Casein kinase II [20,23], the BMP protein encoded by *decapentaplegic* [24,25] and by the calcium-dependent modulatory protease Calpain A [26]. Moreover, Calpain A generates a Cact fragment that is more stable than full-length Cact [17,27], indicating that the Toll-independent pathway may perform unexplored roles in nDl gradient formation. Free Cact seems to be the preferential target for this Toll independent pathway, since at least one of its elements, namely Calpain A, has no effect on Cact that is complexed to Dl [17]. Accordingly, free Cact molecules (not complexed to Dl) have been detected in wild type embryos [17,28].

Recently, by analyzing the effects of a series of *cact* and *dorsal* (*dl*) loss-of-function alleles, we detected a novel function of Cact to promote Dl nuclear translocation. In addition to inhibiting Dl nuclear import, Cact also acts to favor Dl nuclear translocation where Toll signals are high [29]. However, the mechanism behind this effect is unclear. Since previous mathematical models for Dl gradient formation did not distinguish the two mechanisms that regulate Cact function, here we propose a new reaction-diffusion model to describe this signaling network. We take into account non-uniform activation of Toll and add translational regulation of Cact by Dl [30,31]. To distinguish between the Toll-regulated and Toll-independent pathways that control Cact, we add the activation by phosphorylation and degradation by the proteasome terms to the pathway activated by Toll. We also distinguish Dl that translocates to the nucleus in response to Toll versus Dl that flows into nuclei independent of Toll activation. Using a Genetic Algorithm and experimental data from wild-type *Drosophila* embryos, we calibrate the model parameters, including kinetic constants, diffusion coefficients, Toll activation profile, and total Dorsal concentration in the embryo. The optimized parameters are then used to reproduce and understand the nDl patterns of single and double mutants for *cactus* and *dorsal* genes. Our model analysis indicates that, in the ventral region, Cact favors Dl nuclear translocation by replenishing Toll-responsive Dl-Cact complexes that signal to Dl nuclear localization, while throughout the entire embryo Cact binds to and prevents free Dl from entering the nuclei.

## Results

### A reaction-diffusion model that distinguishes two routes for Dorsal nuclear localization highlights the distinctive contribution of Toll-activated versus free Dl flow to the nDl gradient

Embryonically translated from maternal mRNAs deposited in the egg, Dl protein nuclear translocation is regulated at two levels [3,18,30]: 1) High and intermediate levels of nuclear translocation depend on the ventral and lateral activation of Toll receptors, that induce the phosphorylation and proteasomal degradation of the Cacdt inhibitor as well as Dl phosphorylation [32]; 2) Basal levels of Dl flow in and out of all nuclei along the DV axis, as observed by FRAP analysis in dorsal nuclei where Toll signals are absent [18]. Accordingly, Dl dimers have been detected by non-reducing gel electrophoresis of completely dorsalized embryos from *snk-* mutant females, where the Toll ligand is not produced [28]. Immunological detection of

Dl in fixed embryos, or visualization of fluorescently tagged Dl in live tissue, does not discriminate the Dl dimers that enter nuclei in response to Toll versus those that enter by direct flow. In order to investigate how these different Dl nuclear entry modes impact the nuclear Dl (nDl) gradient, we built a model that discriminates between these two routes. In the first route described, activated Toll recruits a 2Dl-Cact trimeric complex (DlC) to the Toll-signaling complex (DlCT) (represented by the reversible reaction related to kinetic constants k7 and k8, Fig 1), leading to the irreversible dissociation of phosphorylated Dl from Cact (k9), Cact degradation (k10), and consequent entry of a Dl dimer in the nucleus (k11). In a final step of this route, the Dl dimer returns to the cytoplasm (k12). The second nuclear entry mode depends on the direct, reversible flow of free Dl dimers in and out of the nucleus (k3 and k4). In the cytoplasm, free Dl dimers can associate to free Cact (k5 and k6), generating DlC that inhibits Dl nuclear translocation. In addition, the model includes Dorsal-mediated translational regulation of Cactus (k1), as well as Toll-independent Cact processing that may generate Cact fragments with different activities and/or leads to Cact degradation (k2) [17,19]. This model reproduces the characteristic ventral-to-dorsal nDl gradient displayed in wild-type embryos during cycle 14, which is required to subdivide the embryo in a series of dorsal-ventral gene expression territories (Fig 2A–2C and Table 1). Furthermore, it allows discriminating the contribution of each Dl nuclear transport entry route along the entire DV axis to generate the compounded nDl pattern (Fig 2D). Thus, a homogenous direct flow of Dl dimers into the nucleus is observed along the DV axis (Fig 2D, $nDl^0$), while Toll-induced nuclear Dl displays ventral-to-dorsal asymmetry (Fig 2D, $nDl^*$). Consequently, in the ventrolateral region, the contribution from the Toll-induced route to nDl levels is much larger than from direct flow, resulting in a significant gradient of nuclear Dorsal. On the other hand, in the dorsal region, the contribution from direct flow is predominant.

Model parameters calibrated to simulate wild-type data. The dimensionless concentrations were obtained by normalizing the experimental data, as described in Materials and Methods. Specifically, for Dl concentration and peak Toll concentration, normalization is obtained by dividing these values by the experimental nuclear Dl concentration in the most ventral region. For quantities involving length, including diffusion coefficients and the width of the Gaussian that represents the Toll activation profile, the total size of the half-embryo was considered to be unitary length (therefore, each compartment is 1/50 in size). For the quantities involving time, such as diffusion coefficients and kinetic constants, the characteristic time was t = 1.

One fundamental characteristic of the embryonic Dl gradient is that it is robust to variations in Dl and Cact levels, tolerating a Dl/Cact ratio of 0.3 to 2.0 without a significant effect on embryo viability [30]. In order to test how the amount of total Dl protein impacts its nuclear localization, we simulated nDl levels in embryos generated by mothers heterozygous for a loss-of-function *dl* allele (*dl*[6], [29]). These embryos (from *dl*[6]/+ mothers) produce less Dl protein than embryos generated from wild-type mothers [29]. Embryos from *dl*[6]/+ mothers display a significant reduction in peak nDl in the ventral domain but show little change in nDl levels in the lateral or dorsal regions (Fig 3A, [11,29]). A subtle widening of the nDl gradient at ventral levels is also observed (S7 Fig). Our model reproduces the reduction in peak nDl and approximates the *dl*[6]/+ pattern in lateral nuclei. In agreement with experimental data showing that Dl controls Cact levels, all Cactus species significantly reduce as compared to control (S1C Fig, free Cact; S1G Fig, ubiquitinated Cact; S1D Fig, DlC and S1F Fig, DlCT). Additionally, our simulations indicate that different Dl components behave distinctively in this context: by lowering the total amount of Dl (Table 2), Toll-activated $nDl^*$ decreases, while $nDl^0$ that enters the nucleus by direct flow does not change significantly compared to wild type (Fig 3B). Accordingly, when the model is tested for greater reductions of maternal Dl (Fig 3D) nDl reduces significantly, particularly in the ventral region where $nDl^*$ is predominant. With

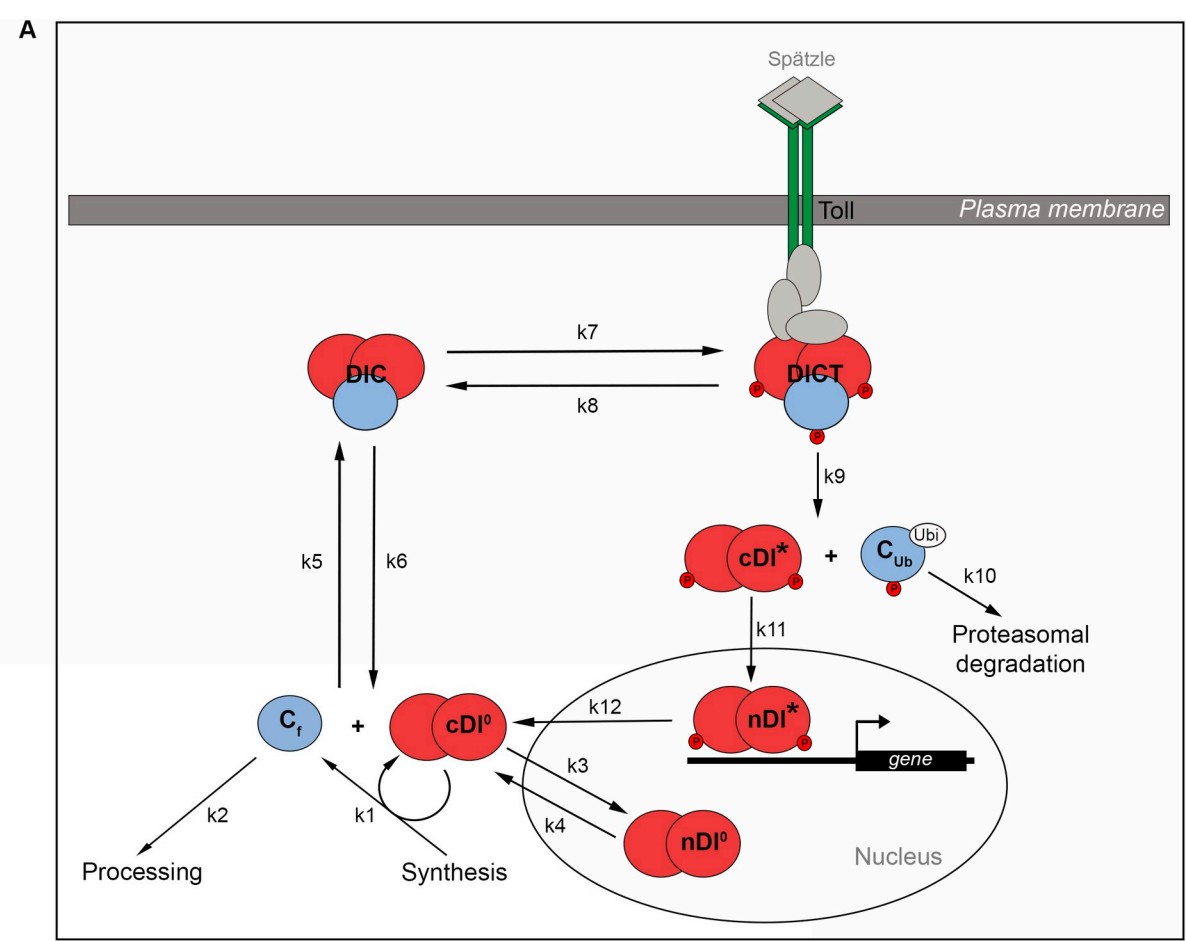

**Fig 1. Mathematical modeling of nuclear Dorsal gradient in *Drosophila* embryos.** (A) Schematic representation of the Reaction-Diffusion Network Model regulating nuclear Dorsal (nDl) localization. Kinetic constants k1 and k2 mediate synthesis and Toll-independent degradation of the inhibitor Cactus, respectively. Dl dimers can enter and leave the nucleus by direct flow (k3 and k4), independent of Toll activation. k5 and k6 mediate reversible binding between cytoplasmic Dl dimers (cDl$^0$) and free Cactus (C$_f$) to form trimeric complexes (DlC). DlCT complex is reversibly formed by the interaction between DlC and activated Toll (T) membrane receptor (k7 and k8). Following DlCT complex formation, Toll activation induces Dl and Cactus phosphorylation, releasing Dl and C from the complex. This irreversible reaction is

controlled by k9. Cytoplasmic phosphorylated Dl dimers (cDl*) enter the nucleus (nDl*) while phosphorylated and ubiquitinated Cactus ($C_{ub}$) is degraded by the proteasome (k10). k12 controls nDl* output from the nucleus. Note that T represents only the activated form of the Toll receptor. (B) Detailed reaction network stoichiometry. (C) Key relationships among model species. Total Cactus (C) is the sum of all species that contain Cactus. Total nuclear Dorsal (nDl) is the sum of $nDl^0$ and nDl*. Total cytoplasmic Dorsal (cDl) includes free cDl, cDl*, plus the two DlC and DlCT complexes. Total Dorsal is the sum of nDl and cDl. The model was solved for cleavage cycle 14 embryos.

a > 20% reduction in Dl, a small decrease of nDl on the dorsal side is also observed, consistent with quantification experiments using a different *dl* loss-of-function allele (*dl*[1]) [9]). Furthermore, the model predicts that the ventral peak of DlCT reduces in this mutant (S1F Fig), while DlC that is uniform along the DV axis, reduces evenly as compared to wild type (Fig 3C). This pattern of DlCT and DlC reduction conforms to the fact that *dl* controls Cact levels (represented by k1): once Dl levels drop, so do Cact levels and the amount of both DlCT and DlC. The reduction in DlCT reduces the concentration of nDl* in the ventral region (where DlCT is predominant). On the other hand, even though the reduction in Dl also decreases the amount of Dl that enters the nucleus by direct flow, the consequent Cact decrease leaves more Dorsal

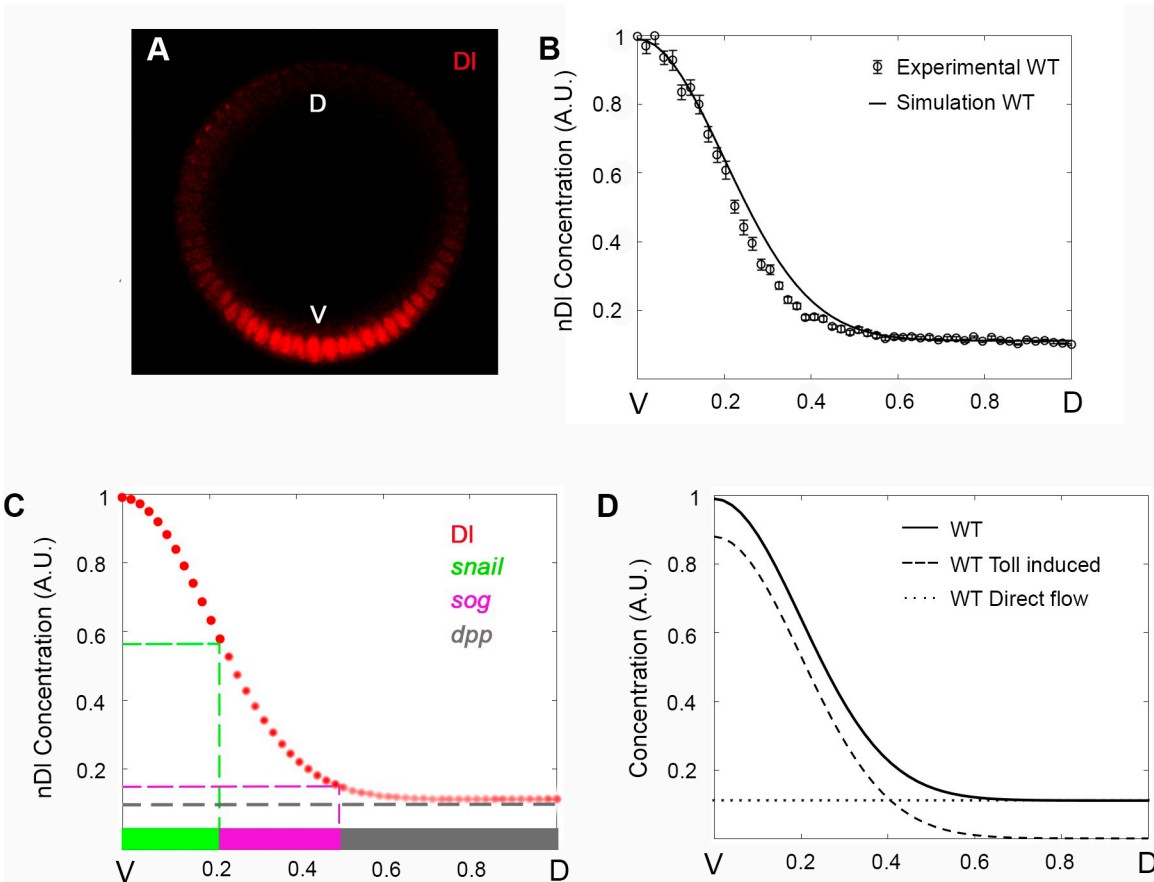

**Fig 2. The Reaction-Diffusion Network Model reproduces the wild type nDl gradient profile and discriminates two different Dl nuclear entry modes.** (A) Optical section of a wild type (WT) embryo stained for Dorsal protein. (B) Nuclear Dl fluorescence intensity was extracted from sections as in A, measured and plotted as half gradients (circle). The black curve displays model simulation. The y-axis represents nDl fluorescence intensity along the ventral-to-dorsal (V-D) embryonic axis (x-axis). Data are mean ± s.e.m. (C) High to low nDl levels (red circles) define different DV territories: ventral mesoderm represented by *snail* expression (green), lateral neuroectoderm represented by *short gastrulation* (*sog*, magenta) and dorsal ectoderm defined by *decapentaplegic* expression (*dpp*, grey). (D) Simulations discriminate nuclear Dl that enters the nucleus by direct flow ($nDl^0$, dotted curve) or induced by Toll (nDl*, dashed curve). The black curve indicates total nDl model simulation, as in B. Ventral (V) region to the left, dorsal (D) to the right.

**Table 1. Dimensionless model parameters.**

| Non-dimensional Parameter | Value | Non-dimensional Parameter | Value |
|---|---|---|---|
| $[Dl]_{tot}$ | 0.525 | $k_4$ | 0.0022 |
| $A_{[T]tot}$ | 8.54 | $k_5$ | 877 |
| $\sigma_{[T]tot}$ | 0.1995 | $k_6$ | 7510 |
| $D_{Dl}$ | 91 | $k_7$ | 0.113 |
| $D_C$ | 415 | $k_8$ | 0.0023 |
| $D_{DlC}$ | $3.39 \times 10{-}5$ | $k_9$ | 0.0036 |
| $k_1$ | 0.584 | $k_{10}$ | 427 |
| $k_2$ | 0.0610 | $k_{11}$ | 1.20 |
| $k_3$ | 0.0043 | $k_{12}$ | 0.0022 |

dimers free for nuclear translocation. Al Asafen *et al.*, 2020 have successfully modeled the pattern of the nDl gradient in the middle of the embryo, focusing on their behavior in response to variations in *dl* expression [33]. They find that by including Toll receptor saturation, presence of nuclear DlC and shuttling of Cactus and Dorsal along the DV axis, their simulations provide

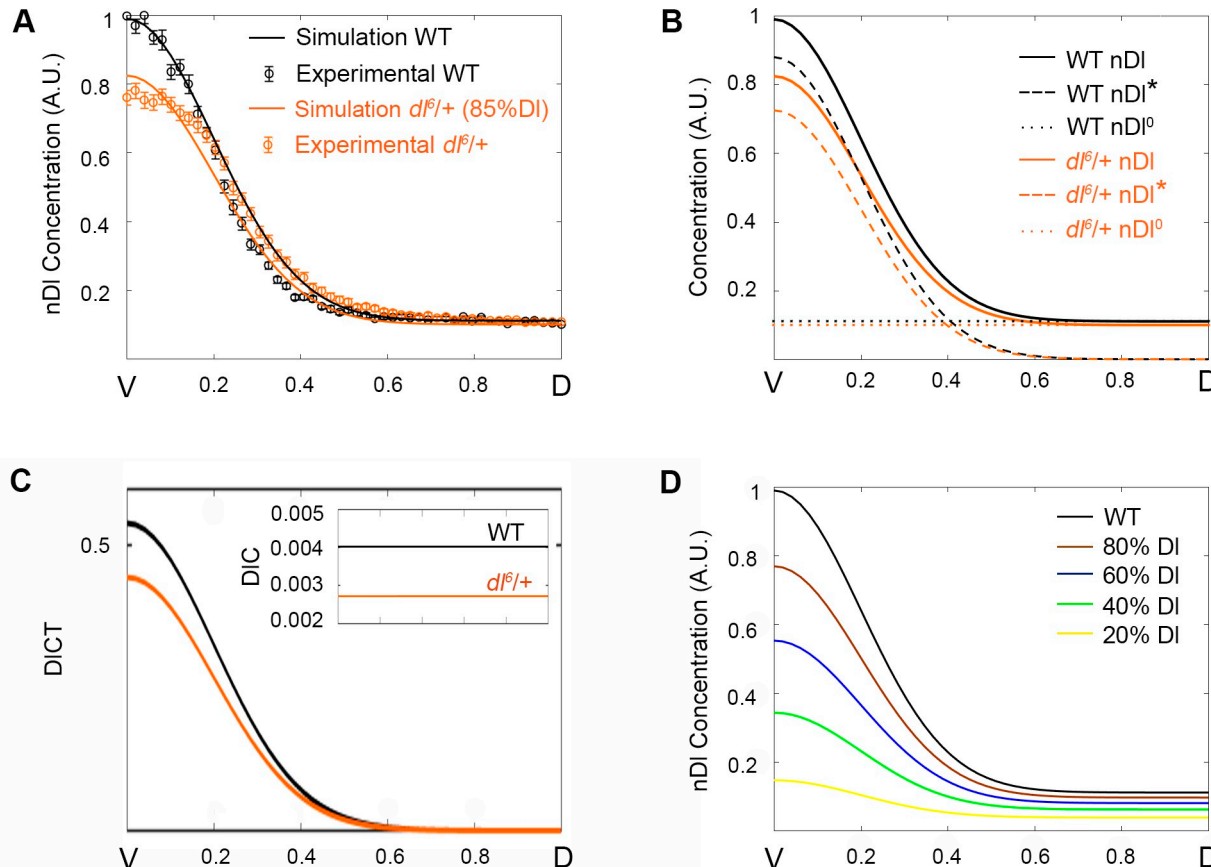

**Fig 3. Dl gradient simulations are responsive to variations in Dl levels.** (A) Simulation (solid curves) and experimental data (circle symbols) from mutant (orange) and wild type (black) embryos were plotted in the same graph. Nuclear Dl gradient from *dl*[6]/+ mutant embryos is simulated and fitted using an 85% reduction in total Dorsal protein level. (B) Spatial distribution of Dorsal protein that enters the nucleus by direct flow (nDl[0], dotted curve), or Toll induced (nDl*, dashed curve) and total nDl. (C) Distribution of DlC and DlCT species amounts for wild type (black) and *dl*[6]/+ (orange) genotypes. (D) nDl gradient simulations resulting from 20% (yellow), 40% (green), 60% (blue), 80% (brown) Dorsal protein reductions compared to a wild type nDl gradient (black).

a good fit for nDl in lateral regions of the embryo, as well as dorsal and ventral nuclei. In the future, it will be interesting to include some of these elements in the model hereby proposed, allowing to investigate their effects in a model that includes different elements of Cactus regulation.

## Cact protein acts antagonistically along the DV axis to control the levels of unbound versus complexed Dl

We have previously shown that Cact displays not only a role as an inhibitor of Dl nuclear translocation, but also acts to favor nDl localization in ventral nuclei (Fig 1 in [29]). In order to use our model to unveil the molecular mechanism behind this dual role exhibited by Cact, we simulated the pattern of hypomorphic *cact* allelic combinations. Initially, we simulated the *cact*[A2]/*cact*[11] genotype, where total Cact levels are reduced in relation to wild type (Table 2), in addition to uncharacterized effects of the mutant alleles [3,17,29]. This genotype shows two opposing effects reproduced by the model: a decrease in nDl in the ventral region and an increase in the lateral and dorsal regions of the embryo (Fig 4A). The nDl increase in lateral and dorsal regions of the embryo reflects Cact's classical role described by many authors to inhibit Dl nuclear translocation [3,6,19,21,30]. Contrarily, the nDl decrease in ventral nuclei, where Toll activation is high, requires a detailed model analysis to be understood.

By analyzing direct and Toll dependent Dl nuclear entry routes separately we found that both are affected, albeit differently, by reducing Cact levels (Fig 4C). The amount of Dl that directly enters the nuclei ($nDl^0$) increases equally along the entire DV axis, consistent with a homogenous increase of free Dl dimers in the cytoplasm ($cDl^0$, S2A Fig). This behavior reflects the well-characterized role of Cact to inhibit Dl nuclear translocation. Conversely, the amount of Dl entering the nucleus in response to the Toll pathway ($nDl^*$) reduces compared to wild type, particularly in the ventral side of the embryo. This $nDl^*$ reduction in *cact*[A2]/*cact*[11] reflects the loss of a positive role of Cact on Dl nuclear localization. The transition between the positive and negative effects of Cact reduction on Dl nuclear levels is placed exactly at the position along the DV axis where Dl translocation induced by the Toll pathway (dashed blue line in Fig 4C) matches the amount of Dl that enters by direct flow (dotted blue line in Fig 4C). This falls inside the lateral domain where intermediate Toll activation takes place. It is important to point out that Dl nuclear import through the Toll pathway is more efficient than via direct flow: the ratio between kinetic constants for Toll-dependent Dl nuclear transport ($k11/k12 = 545.45$) is higher than via direct flow ($k3/k4 = 1.95$; Table 1). This is critical to the dual effect observed for *cact*[A2]/*cact*[011]. Since the decrease of Dl that translocate to the nucleus in response to Toll ($nDl^*$; Fig 4C) is greater than the increase in Dl entering the nuclei by direct flow ($nDl^0$; Fig 4C), the resulting effect is a reduction of total nuclear Dl ($nDl^0 + nDl^*$) in the most ventral region (Fig 4B and 4C). Importantly, in addition to the difference in nuclear resident time between $nDl^0$ and $nDl^*$ suggested by the ratios above, we

**Table 2. Dimensionless parameters for each mutant.**

| Genotypes | [Dl]tot | k1 | [C]tot |
|---|---|---|---|
| WT | 0.525 | 0.584 | 100% |
| *dl*[6]/+ | 0.4605 | 0.584 | 90.70% |
| *cact*[A2]/*cact*[011] | 0.525 | 0.1825 | 53.40% |
| *dl*[6]/*cact*[A2] | 0.375 | 0.500 | 71.80% |

Parameters obtained for each mutant simulated, compared to wild type. Only the changeable values are presented, as all other parameters of the model were kept fixed.

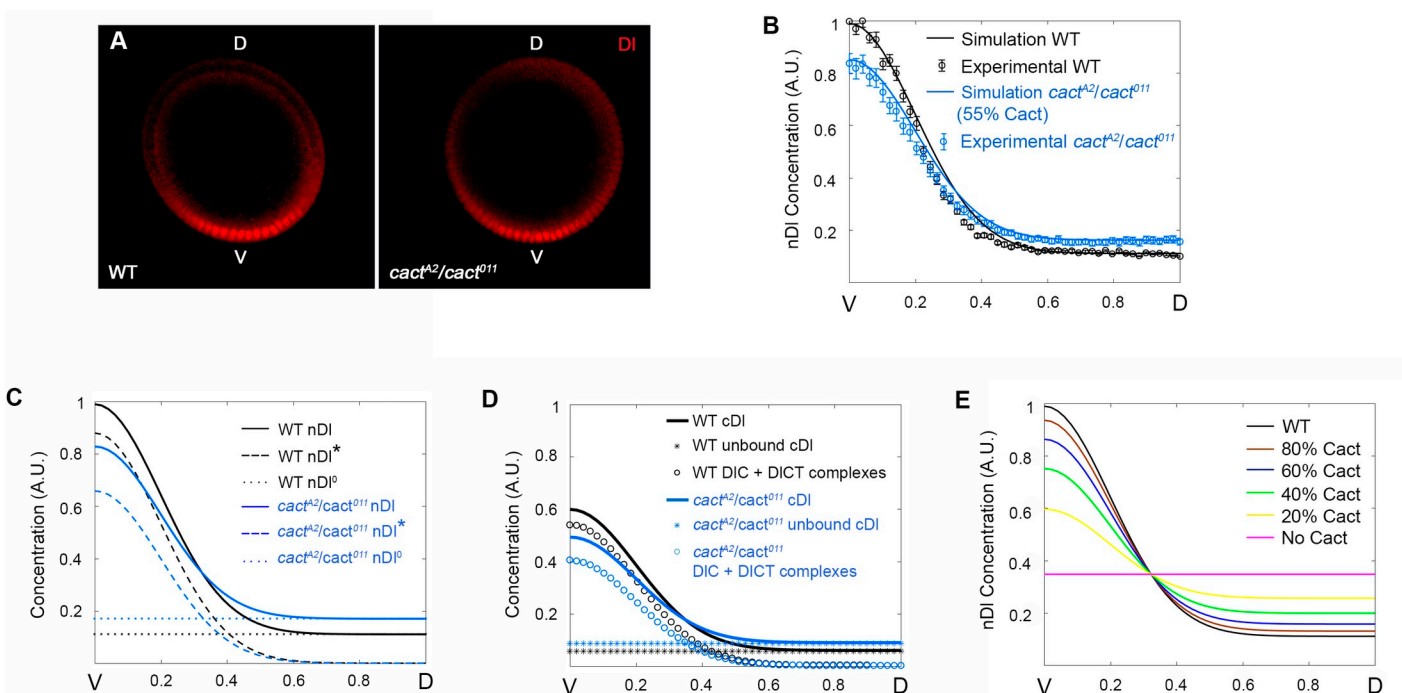

**Fig 4. Cact produces distinct effects along the DV axis by controlling unbound versus complexed Dl elements.** (A) Simulations (solid curve) and experimental data (circle symbols) from *cactus* mutant (blue) and wild type (black) embryos. Nuclear Dl gradient from $cact^{A2}/cact^{011}$ mutant embryos are simulated and fitted using a 55% reduction in Cact protein. (B) Spatial distribution of Dorsal that enters the nucleus by direct flow ($nDl^0$, dotted curve), by Toll induced ($nDl^*$, dashed curve) and total nDl. (C) Distribution of Total cDl (defined as $cDl^0 + DlC + DlCT + cDl^*$; star symbols), sum of the unbound cDl species ($cDl^0 + cDl^*$) and the DlC + DlCT complexes for wild type (black) and $cact^{A2}/cact^{011}$ (blue) genotypes. (D) nDl gradient simulations resulting from 100% (magenta, constant nDl), 20% (yellow), 40% (green), 60% (blue), 80% (brown) Cactus protein reductions are compared to wild type nDl gradient (black).

should note that the effect of $nDl^0$ and $nDl^*$ on gene expression will also depend on their distinct efficiencies to bind target DNA.

Interestingly, if we consider the different cytoplasmic Dl species that are free ($cDl^0 + cDl^*$) versus bound to Cact (DlC + DlCT; Fig 4D), we see that the amount of Cact-bound Dl decreases in the *cact* mutant as compared to wild type. This conforms to experimental data from crude embryonic lysates showing that in *cact* loss-of-function alleles the total amount of Dl-Cact complexes observed in non-reducing gels decreases compared to wild type [28]. Accordingly, in our analysis, as the amount of DlCT (S2F Fig) reduces, a consequent reduction in the amount of modified Dl in the cytoplasm ($cDl^*$) and nuclei ($nDl^*$) is seen (S2H and S2I Fig). This $nDl^*$ decrease is in agreement with the loss of ventral gene expression (high Dl activated genes) and ventral expansion of lateral gene expression (intermediate Dl level targets) that is observed in this genotype [29]. Therefore, DlC and DlCT complexes are critical to the dual effect observed in *cact-* mutants: with the decrease in DlC levels, compared to wild type, the inhibitory effect of Cact is reduced, especially in dorsal regions of the embryo; the decrease in DlCT complexes in Toll activated regions, impairs Dl nuclear translocation ventrally and laterally.

In order to investigate the effect of more severe reductions in Cact concentration on the amount of Dl that enters the nuclei, we performed additional simulations with different production rates of Cact, from wild-type rates to zero, that is, in the absence of Cact (Fig 4E). The model predicts an increase in the Cact dual effect resulting from reductions in Cact levels: In all conditions a more severe loss of Cact lead to less nDl in the ventral side and more nDl in dorsal regions, as compared to wild-type embryos. Furthermore, the model correctly predicts

that loss of Cactus function progressively leads to flattening of the Dorsal gradient, reflecting the reported pattern of *cact-* embryos with expanded lateral territories [29,34–36]. On the other hand, germline clones for loss-of-function *cact* alleles, which more closely approximate the zero Cactus condition, still retain some polarity, suggesting that other processes that are not reflected in our model are involved in gradient formation. One such process could be Dl phosphorylation [32], which has been shown to increase nuclear intake and DNA binding, likely decreasing the amount of free Dl that flows out of the nucleus. Although our model incorporates Dorsal phosphorylation in response to Toll, it does not allow Dl phosphorylation in *cactus* absence. Additional experimental data is required to understand in full the effects of Dl phosphorylation and to allow the incorporation of these reactions in future mathematical models.

### *dl/cact* double mutant reduces the levels of Toll-responsive Dorsal-Cactus complexes causing loss of precision in target gene expression domains

Since our results indicate that Cact favors Dl nuclear translocation in the ventral side of the embryo, we decided to challenge this prediction by decreasing Dl and Cact concomitantly. If the model prediction is correct, reducing both Cact and Dl should produce a more severe impact in the ventral nDl peak compared to the single *dl*[6]/+ mutant. This condition is experimentally reached with the *dl*[6]/*cact*[A2] allelic combination, where the levels of both Dl and Cact are reduced (Figs 5A and S3). We simulated the *dl*[6]/*cact*[A2] mutant by decreasing total Dl (maintained constant throughout the simulation) and the kinetic constant k1, related to Cact production (see Fig 1A and 1B and Table 2). The model shows that lowering Cact levels in a *dl*[6]/+ background produces an additional impact on Dl nuclear translocation in the most ventral region and confirms that Cact has a positive role on Dl nuclear localization. Model discrimination of Toll-induced (nDl*) and direct Dl flow (nDl[0], Fig 5B) indicates that in this allelic combination both Dl nuclear translocation mechanisms are reduced.

In the dorsal region, the effect of reducing Cact production (that has the potential to increase nDl levels, Fig 4) is almost completely canceled by the reduction in Dl levels (Fig 5A). Consequently, the dual effect of Cact reduction is no longer observed (compare Figs 5A and 4B). By distinguishing the amount of nDl* and nDl[0], the model shows that nDl resulting from direct flow (nDl[0]) decreases uniformly along the DV axis and that Toll-responsive nDl* decreases ventrolaterally (Fig 5B). Therefore, in ventral regions, the reduction in nDl is

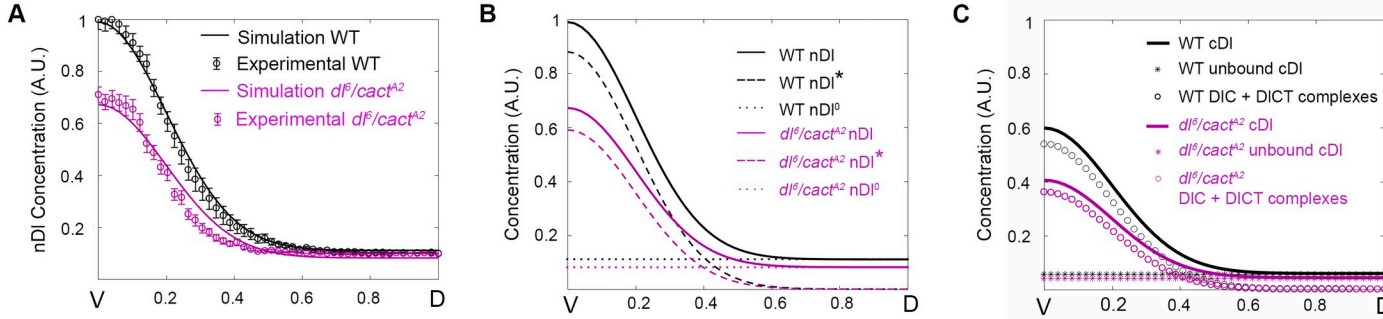

**Fig 5. A *dl/cact* double mutant highlights the positive role of Cact on Dl nuclear translocation.** (A) Simulations (solid curves) and experimental data (circle symbols) from mutant (purple) and wild type (black) embryos. Nuclear Dl gradients from *dl*[6]/*cact*[A2] mutant embryos were simulated using simultaneous reduction of 50% Cactus and 70% Dorsal. (B) Spatial distribution of Dorsal that enters the nucleus by direct flow (nDl[0], dotted curve), by Toll induced (nDl*, dashed curve) and total nDl are shown in wild type (black) and mutant (purple) simulations. (C) Distribution of Total cDl (defined as cDl[0] + DlC + DlCT + cDl*; star symbols), sum of the unbound cDl species (cDl[0] + cDl*) and the DlC + DlCT complexes for wild type (black) and *dl*[6]/*cact*[A2] (purple) genotypes.

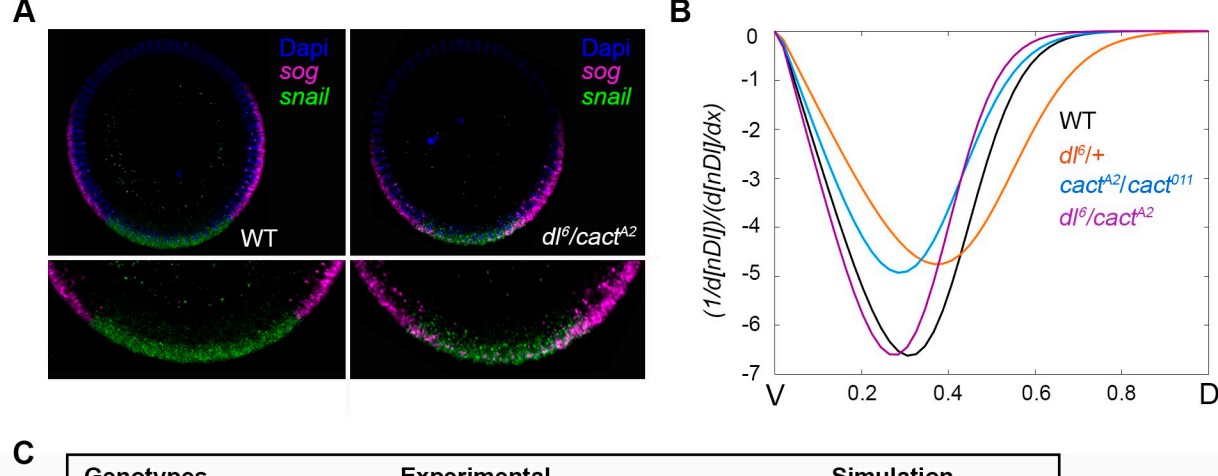

**Fig 6. Variations in nDl gradient slope disrupts the precision of target gene expression DV territories.** (A) Cross-sections of wild type (WT) and mutant (*dl⁶/cact^A2*) cleavage cycle 14 embryos hybridized with *snail* (green) and *sog* (magenta) antisense RNA probes. Nuclei were stained using DAPI (blue). Note *sog* transcripts invading the ventral territory in *dl⁶/cact^A2* mutants (zoomed images). (B) Derivatives of nDl concentration (y-axis) as a function of the position along the embryonic DV axis (x-axis) for WT (black), *dl⁶/+* (orange), *dl⁶/cact^A2* (purple), and *cact^A2/cact^011* (blue) mutant embryos. (C) Peak, basal levels, amplitude and highest slope of nDl distribution for each different genetic background.

compounded by a decrease in both Dl translocation by Toll-dependent and direct flow entry modes. Importantly, the effect on the amount of complexed (DlC + DlCT) versus free cytoplasmic Dl observed in *dl*[6]/*cact*[A2] (Fig 5C) contrasts from the effect of reducing Cact alone (Fig 4D). In *dl*[6]/*cact*[A2], both complexed and free Dl are reduced compared to wild type, while in *cact*[A2]/*cact*[11] a reduction in Dl complexes and increase in free Dl is observed. This indicates that a decrease in the relative amount of complexed versus free Dl is critical for the dual effect observed for *cact* loss-of-function.

In considering the role of a morphogen and how the regulatory network impacts morphogen activity, it is important to address how the network affects target gene expression domains (see Fig 2C). A current discussion in the literature concerns whether changes in the slope of the Dl gradient affect target gene expression, especially in the lateral region [9,10]. It is discussed whether differences in the relative amount of nDl in neighboring nuclei are sufficient to define the necessary thresholds for differential gene expression. In wild-type embryos, the *snail/sog* sharp boundary delimits the ventral mesoderm and lateral neuroectoderm domains, positioned at 20% of the DV axis. In *dl*[6]/*cact*[A2] embryos, which lack ventral-lateral boundary precision, the border between the *sog* and *sna* domains is not clearly defined, and *sog* exhibits stochastic expression invading the ventral region (Fig 6A; [29]). Therefore, we decided to investigate whether the Dl gradient slope was different among all the genotypes hereby analyzed. To this end, we plotted the derivative of the total nuclear Dorsal concentration (nDl) with respect to the DV axis (Fig 6B), where the derivative represents the difference in nDl levels between neighboring nuclei along the DV axis. We observed that, although all nDl gradients show similar shape, the highest slope values are different for each genotype. Indeed, wild type achieves the greatest absolute slope, whereas the gradient displayed by the *dl*[6]/*cact*[A2]

genotype shows the lowest slope (Fig 6C). This result suggests that in wild-type embryos, the lateral adjacent nuclei exhibit a greater difference in the relative amount of nDl than *dl*[6]/*cact*[A2], which could lead to a precise definition of the embryo's domains in the control, while the mutant displays less precise domains. This analysis indicates that the relative concentration of nuclear Dl between neighboring nuclei indeed plays a role in domain precision. Conversely, loss of precision in the *sna/sog* border is not observed in *dl*[6]/+ and *cact*[A2]/*cact*[11] embryos [29], indicating that these two mutants still exhibit sufficiently large slopes to define nuclear interdomain differences. Importantly, it has been shown that the refinement of ventral-lateral gene expression borders is also mediated by RNA polymerase II stalling on transcription start sites [37–39], as reported for *short gastrulation (sog)* [37]. Therefore, a combined effect of distinct nDl levels and transcriptional processes may define the precise gene expression border between adjacent ventral and lateral nuclei of the Drosophila blastoderm embryo.

## Model simulations reproduce Toll-dependent and Toll independent pathway mutant conditions

To challenge the predictive strength of our model, we simulated different mutant conditions described in the literature and their effects on the nDl gradient. First, we simulated the blockage of intracellular signals that lead to Cactus degradation, as in a *pelle* mutant (*pll-*). Pelle is a DEAD-domain kinase activated downstream of the Toll receptor, likely involved in Cact phosphorylation with consequent ubiquitination and degradation by the proteasome [20,40]. Loss-of-function maternal *pll* mutants lead to dorsalized embryos, with loss of lateral and ventral elements of the cuticle, phenotypically akin to loss-of-function Toll mutants [13,41]. By reducing k9 or k10, the constants controlling Cact phosphorylation and release of Dl dimers for nuclear translocation (Fig 1), nDl levels gradually decrease. At k9 and k10 = 0, all nuclei display the same level of nDl, equivalent to levels observed in the dorsal region of wild-type embryos (Fig 7A). Accordingly, the amount of DlCT and DlC accumulates (Fig 7B and 7C), since Dl dimers are not released from Cact inhibition. This effect is restricted to ventral and lateral regions where pre-signaling complexes are recruited to active Toll. No difference from the wild type condition is seen in the dorsal region, where nDl results solely from direct Dl flow.

Next, we simulated a decrease in activated Toll (Fig 7D). In this simulation, decreasing activated Toll ($Tl^{act}$) leads to less efficient interaction with the DlC complex, as compared to wild type. As a result the nDl gradient progressively flattens, conforming to the reported loss of DV polarity that results from loss of Toll [34]. Accordingly, DlCT reduces while DlC increases progressively compared to wild type (Fig 7E and 7F). Surprisingly, however, once Toll interaction with DlC is completely impaired (Fig 7D, inactivated Toll, $Tl^{in}$), intermediate and uniform levels of nDl are observed in all nuclei, above the concentration expected for a dorsalized Toll null phenotype [35,36]. A closer look at the levels of the different species that contain Dl could explain this phenotype. Contrary to the *pll-* simulation (Fig 7A), where DlCT is high and keeps all Dl from entering the nucleus, loss of $Tl^{act}$ (Fig 7D) leads to high DlC. Therefore, at $Tl^{act} = 0$ the resulting flat nDl pattern is composed solely of $nDl^0$, which is less efficient in activating target gene expression than $nDl^*$ [32]. Thus, an apolar embryo with a dorsal-lateral gene expression pattern is expected, which more closely resembles the non-polar and dorsalized *Tl-* phenotype.

We also changed kinetic parameters that are not directly related to Toll receptor activation, namely, k2 and k5, to explore their effects on Cact and Dl distribution. Decreasing these constants led to contrasting effects on dorsal versus ventral regions of the embryo, replicating the dual effect we observe in *cact*[A2]/*cact*[11] (S5–S5H Fig). In addition, by increasing k5 we were able to recover the *cact*[A2]/*cact*[11] dual phenotype (Fig 7G–7I). A similar effect is

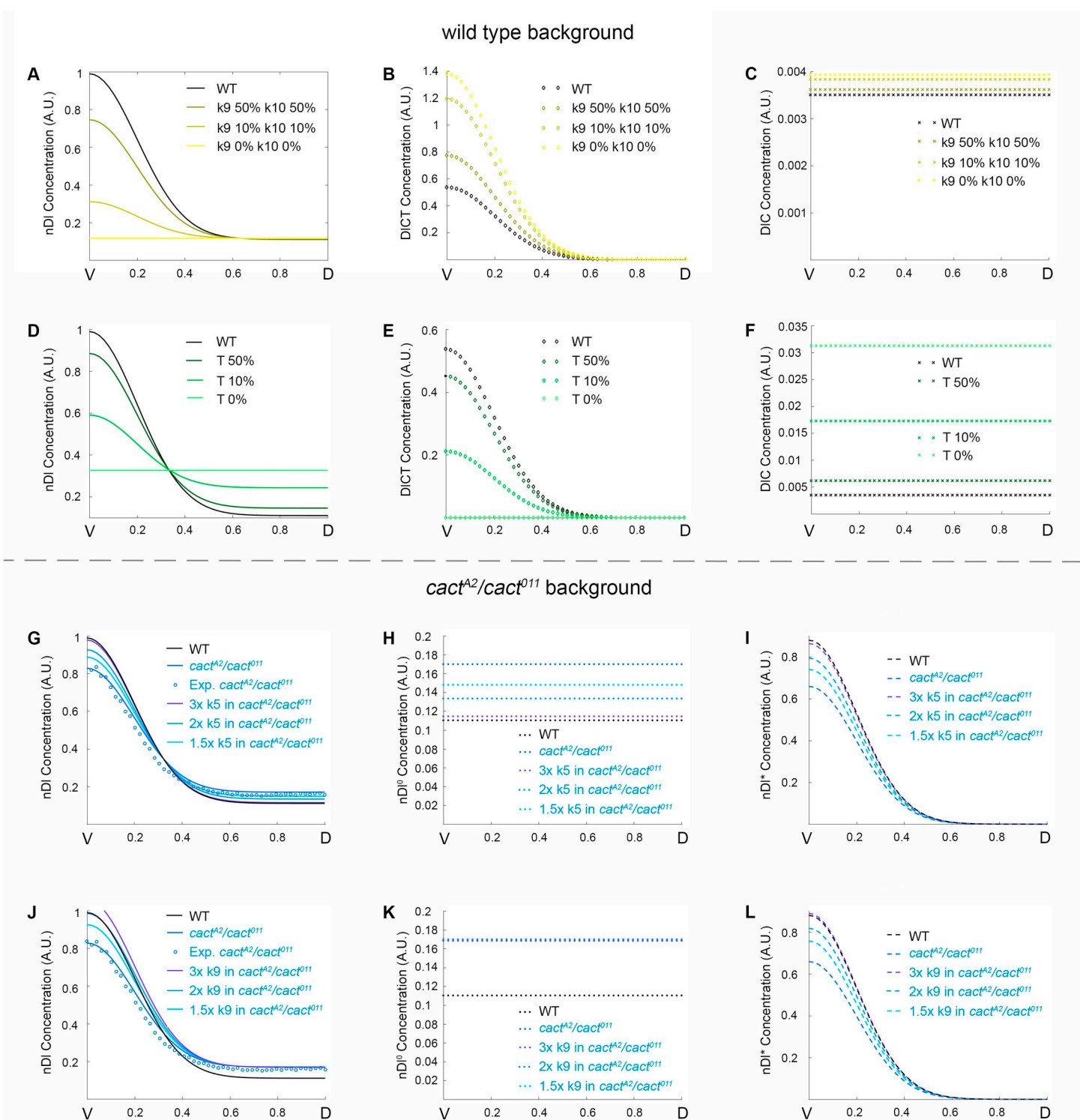

**Fig 7. Model simulations reproduce Toll-dependent and Toll independent pathway mutant conditions.** (A-C) Simulations for simultaneous reduction of kinetic constants k9 and k10 (0%, 10% and 50% reductions), showing the distribution of nuclear Dorsal (A), DlCT complex (B) and DlC complex (C). (D-F) Simulations decreasing activated Toll receptor ($Tl^{act}$) to 10% or 50% or inactivated Toll ($Tl^{in}$), showing the distribution of nuclear Dorsal (D), DlCT complex (E) and DlC complex (F). (G-L) Simulations increasing kinetic constant k9 (G-I) or k5 (J-L), by 3x, 2x or 1.5x, in a $cact^{A2}/cact^{011}$ mutant background. Shown are nDl (total nuclear Dorsal dimers, G, J), nDl⁰ (free nuclear Dorsal dimers; H, K) and nDl* (nuclear Dorsal dimers activated by Toll Pathway; I, L). G and J also display experimental data from the $cact^{A2}/cact^{011}$ mutant (circles) for comparison with wild type and mutant simulations. All concentrations are plotted along the V-D axis.

observed by increasing k2 or k7 in the same *cact*[A2]/*cact*[11] background (S5A–S5D Fig). Perfect recovery to the wild-type pattern is paralleled by a homogeneous decrease in nDl$^0$ (Figs 7H and S5B) and ventral increase of nDl* (Figs 7I and S5C). Interestingly, this effect closely resembles that seen by providing excess Cact[E10]eGFP to *cact*[A2]/*cact*[11] embryos [29]. This suggests that the N-terminal deleted Cact[E10] fragment inhibits basal nuclear translocation of Dl (nDl$^0$) and favors nDl* by an unknown mechanism that should be explored in the future. We also simulated the effect of enhancing Toll signals in a *cact*[A2]/*cact*[11] background, by increasing k9. As a result, in ventral regions of the embryo nDl recovers the simulated *cact*[A2]/*cact*[11] gradients to wild-type levels. However, the gradient is unchanged in dorsal regions (Fig 7J–7L). Altogether, these results indicate the dual effect observed in *cact* mutants requires Cactus regulation by events that are not a direct result of Toll receptor activation.

## Discussion

### Novel elements incorporated into the regulatory network that controls nDl gradient formation

The *Drosophila* embryo is a unique setting for investigating Toll pathway function, as each genetically identical nucleus receives a unique level of Toll activation, distinct from its neighbor nuclei. Dorsal-ventral patterning of the embryo depends on this graded activity, instigating scientists to model the steps from Toll activation to formation of the nuclear Dorsal gradient. By focusing on the contribution of Cactus molecules, the predictive model herein proposed adds novel elements to regulatory network modeling of nDl gradient formation in the *Drosophila* embryo and expands our understanding of mechanisms that control NFκB/c-Rel activity.

The basis of the model we propose incorporates several elements that were already present in Kanodia *et al.*, 2009 [8]. However, previous models assumed general degradation of free Cact. As Toll-independent free Cactus degradation has been proven experimentally [6,17,19,21], additional reactions were required to explore the effects of degrading Cact in response to Toll versus those induced by Toll-independent mechanisms. To distinguish between Toll-activated and Toll-independent regulation we have included in our model a series of experimentally determined elements and processes. These are: 1) We have introduced Cact phosphorylation and ubiquitination (k9 and k10) in response to Toll activation [6,21,41]; 2) As a consequence, in the current model, free Cactus degradation (k2) can take place in the absence of Toll activation, although the terms free Cactus ($C_f$) and free Cactus degradation were already present in previous models; 3) Dorsal-dependent Cactus protein synthesis (k1). Genetic and biochemical analysis has shown that Dl regulates Cactus levels [6] and likely involves the control of *cact* mRNA stability or translation [31]; 4) Addition of an activated form of Dorsal (Dl*) that is generated in response to Toll allows us to distinguish it from a pool of Dl bound to Cact that is not yet part of a Toll pre-signaling complex. Cytoplasmic binding of free Cactus to non-phosphorylated Dl to build new 2DlC complexes (k5, k6) is maintained as represented in previous models [8,11]. Building Dl-C association-dissociation events and Toll-dependent dissociation as parallel pathways is based on the knowledge that Dl-Cact complexes are observed in Toll pathway loss-of-function mutants [28] and that free Cact, that binds Dl to form new DlC complexes, is controlled by signals independent of Toll [6,17,20,21,26,28].

All published models include the recruitment of DlC to signaling complexes, which is dependent on Toll activation (herein referred to as reactions k7, k8), as well as downstream steps to Toll-dependent Dl-C dissociation, Dl nuclear translocation and degradation of Cact

through the proteasome. The direct flow of Dl in and out of the nucleus (k3, k4) was previously included in other models, either independently or as part of a complex with Cact [11,33,42].

By incorporating the novel Toll independent elements described above our model enables, among other things, to explore the contribution of Toll-activated versus direct Dl nuclear translocation to the nDl gradient. In simulating a wild-type context, the model predicts a uniform level of nDl imported by the direct pathway, consistent with the previously reported uniform flow of dimers from cytoplasm to nucleus along the DV axis [18]. According to model parameters, Dl nuclear translocation by the Toll pathway is more efficient than by direct flow, which is supported by experimental data [32]. Furthermore, by predicting the individual concentrations of all chemical species, the model indicates that the majority of cytoplasmic Dl is bound to either Cact and/or Toll receptor (DlC and DlCT), conforming to the relative levels of free versus Cact-bound Dl protein in embryonic extracts [28].

Our model indicates that the direct flow of Dorsal is an important process to define the shape of the nDl gradient in control and mutant simulations, and accounts for the basal nDl levels experimentally detected on the dorsal side of the embryo at nuclear cycle 14. The lateralized gene expression pattern of double *cact-;Toll-* or *ndl-;cact-* embryos indicates that free Dl dimers are capable of entering the nucleus to regulate gene expression [3,36]. This $Dl^0$ pool likely displays slower nuclear import and lower activity as a transcription factor than Toll-activated Dl ($Dl^*$), since Dl phosphorylation in response to Toll increases nuclear import [32] and transcriptional activity [43], characteristics that should be included in future extensions of the current model. Based on live imaging of Dl-Venus, mathematical modeling by O'Connell and Reeves (2015) suggests that the nuclear flow of a Dl-Cact complex (nDlC) may account for basal nDl staining [42]. Since Cactus has been observed in the nucleus of other Drosophila tissues [44,45], it is plausible that it enters embryonic nuclei to favor Dl export, as shown for vertebrate IκB [46,47]. Both our model and that proposed by O'Connell and Reeves (2015) agree that two different pools of nuclear Dl exist [42], with distinct transcriptional properties from Toll-activated Dl dimers, and that inclusion of these two pools allows a better fit of the models to experimental data. Future experiments addressing the nuclear localization of Cact and Dl complexes shall help to define whether nuclear protein complexes such as $2Dl^0$ and 2DlC coexist in the nucleus and their respective contribution to the nuclear Dl gradient.

## Cactus plays a positive role to induce Dl nuclear translocation by controlling the levels of Dorsal-Cactus complexes

One of the main goals of our work, the proposed model reproduces the dual effect of *cact* loss-of-function combinations and points out the essential components of the network that enable this dual effect. Broadly accepted as a NFκB family protein inhibitor with the role of securing Dl in the cytoplasm, our finding of a Cact function to favor Toll signals implies the existence of unexplored mechanisms that control Dl activity. Our simulations implicate the dynamics of DlC and DlCT complex formation as essential steps for the dorsal increase and ventral decrease of nDl in *cact-* mutants. Importantly, in our model, the generation of DlC also depends on free non-phosphorylated Dl that flows out of the nucleus independent of Toll.

Our model suggests that, by forming DlC, Cact plays a negative role to impair $Dl^0$ from entering the nucleus, while by enabling new DlCT complex formation in response to Toll activation it favors $Dl^*$ nuclear translocation. Since Toll activity is high ventrally, DlCT predominates in this region. Contrarily, in the dorsal region, only DlC is observed. Therefore, simulations suggest that this differential distribution, together with the relative balance between the opposite roles that Cact plays in each DV region, are the characteristics that enable Cact dual effect on Dl nuclear translocation (Fig 4). The generation of endogenous *dl* and *cact*

alleles that either increase or decrease the rate of association between Dl, Cact and Toll pathway adaptor proteins should enable to test these predictions by altering the relative amount of free protein versus protein complexes.

By simulating additional mutant conditions that decrease Cact function, our model confirms that DlC formation from free Cact and $Dl^0$ are important elements for the dual effect of Cact to favor Dl nuclear translocation ventrally and inhibit Dl nuclear translocation dorsally. These simulations indicate that the lower the Cact levels, the greater the amount of nDl that enters the nucleus by direct flow, contributing to increasing the basal (uniform) component of nDl concentration. Notably, increasing Toll pathway activity (increasing k9) does not recover the full *cact* loss-of-function phenotype. Only by modifying the constants associated with Toll-independent processes we were able to recover the phenotype to a wild-type pattern. This indicates that the Toll-independent pathway is fundamental for Cact to perform its different effects.

## Two routes for Cact regulation as an extension of previous models

The pioneer model proposed by Kanodia *et al.* (2009) to explain Dorsal gradient dynamics during *Drosophila* embryogenesis was very successful in pinpointing the most important features of gradient formation during nuclear division cycles 11–14 [8]. This model also proved extremely powerful when comparing nDl gradient formation in related *drosophilids* [48]. In this publication, Ambrosi *et al.* also suggest that Cactus is the most important element that allows the correct gradient accommodation in embryos of greatly divergent sizes. However, previous models confer Cactus only an inhibitory role, where low Cactus levels in the cytoplasm are expected to decrease Cactus-Dorsal association, favoring Toll dependent Dorsal nuclear localization. Inspired by the fact that in certain Cactus loss-of-function mutants nuclear Dorsal concentration decreases in ventral regions, we have proposed a two-step model that accounts for this peculiarity. First, the concentration of Cactus-Dorsal complex is controlled by a Toll-independent mechanism involving Cactus synthesis and processing or recruitment into a complex with Dl. Second, the DlC complex associates with Toll, which promotes facilitated entry of Dorsal into nuclei and irreversible Toll-dependent Cactus degradation. In this case, lower Cactus levels result in a reduction of DlC + DlCT complexes compared to wild type and consequently, contrary to general knowledge, a decrease in Toll-induced Dorsal entry into the nuclei whenever the balance between Toll signals and availability of new DlC + DlCT signaling complexes is perturbed. The general result of this model is that nuclear Dorsal follows the Toll activation gradient plus a basal uniform concentration due to the direct entry of Dorsal independent of Toll.

A fundamental feature of our network that enables to simulate Cactus' positive function is that the formation of DlC complexes depends not only on Toll activation but also on association-dissociation events that may be controlled by Toll-independent inputs. Variation in the amount of DlC complexes may explain why the free flow of Dorsal into the nucleus ($nDl^0$) is almost unaffected in *dl[6]*/+ mutants, compared to the significant decrease in Toll-induced nDl ($nDl^*$). DlC decreases roughly 30% as a result of a 15% decrease in total Dl protein (Fig 3C, inset), whereas free Dl reduces only 10% ($cDl^0$ and $nDl^0$; S1 Fig) as there is less Cact to form DlC than in the wild-type condition. On the other hand, in the ventral side of the embryo 20% less nDl enters the nucleus in response to Toll activation ($nDl^*$) since there is less DlC to form new Toll responsive complexes (DlCT) than in wild-type embryos. Accordingly, in the ventral domain of *cact*[A2]/*cact*[11] embryos, DlC decreases 50%, leading to a great decrease in DlCT and Toll-induced Dl nuclear translocation, although direct Dl flow increases (S2 and S4 Figs). It is noteworthy that, since the ratio between kinetic constants for Toll-dependent Dl

nuclear transport (k11/k12 = 545.45) is higher than via direct flow (k3/k4 = 1.95), the Toll induced Dl nuclear translocation process is more efficient than Dl free flow. Therefore, alterations in DlC affect differently the two Dl nuclear translocation routes.

The observation that DlC is central to explaining the behavior of the different mutant conditions hereby analyzed points to the importance of identifying regulators of DlC formation. Processes that control the displacement of DlC to mount new Toll-signaling complexes (DlCT) or that alter the availability of Dl and Cact for association-dissociation events are likely to induce unanticipated effects on NFκB family transcription factor activity. Among the signals that control Cactus independent of Toll, Calpain A, which impacts the nDl gradient [17], may also control the formation of DlC and DlCT complexes since it localizes close to the membrane where these events should take place [26]. Future investigations on the effect of Calpain A and other Toll-independent pathway regulators on DlC levels and free Dl and Cact may shed light on these important issues.

While building a new model centered on Cactus regulation, some elements included by others have been initially left out. For instance, our model does not incorporate Toll receptor saturation at lateral positions of the embryo, nor does it include the shuttling of DlC complexes towards the ventral side of the embryo as in [11]. It has been suggested that shuttling by Cact is required to concentrate high levels of Dl in ventral regions, and responsible for the widening of the nDl gradient observed in *dl*/+ [11,33,42]. In our assays, by quantifying the Dl gradient at the anterior tip of the embryo, we do observe a subtle widening of the nDl gradient that is not reproduced in our simulations. Additionally, our model could be improved to attain a better fit in lateral positions of all embryos, wild type and mutant. On the other hand, without taking into account shuttling or Toll saturation our model does reproduce the overall Cact dual behavior, suggesting that the main elements for this novel function to favor the nDl gradient are in place. Importantly, the formation of new DlC complexes by a Toll-independent route herein proposed does not dismiss processes such as shuttling of DlC complexes along the DV axis as suggested by others [11,33]. In fact, the movement of a morphogen along a cellular field is currently suggested to involve interactions with different molecules. Considering a cytoplasmic morphogen such as Dl, these could include interactions with scaffolding proteins and proteins that regulate cytoplasmic complex formation themselves. It will be interesting to incorporate additional elements from each model to challenge the contribution of DlC complex formation versus shuttling mechanisms and test their impact on the nDl pattern.

## The balance between reaction and diffusion of free and complexed Dorsal and Cactus

Molecular diffusion of proteins involved in the response to Toll signals greatly impacts the nuclear Dorsal concentration gradient. In order to analyze the diffusion related results of our models, first, we determine its time scale. To this end, we calculated the effective kinetic constants of nuclear Dl import (k3 and k11) and nuclear Dl export (k4 and k12) using concentration-weighted kinetic constants average, and found, respectively, 0.0373 and 0.0022 in inverse of time units. Using the time scale of Dl nuclear translocation found by Carrell *et al.*, 2017 [11], in the order of 2 to 5 min, it gives an estimate of 4 to 11 seconds for the typical time scale of our model (more details in Materials and Methods). It results in a diffusion coefficient for the Dorsal-Cactus complex (DlC) within those compatible with the biological system time scale [11], ranging from 0.2μm$^2$/s to 0.5μm$^2$/s. Moreover, it is noteworthy that $D_{DlC}$ / $L^2 \approx$ 0.08475 is much lower than k11, which favors the entry of Dl into the nucleus via the Toll pathway in the ventral and lateral regions. This value is also much lower than k6, the reversible

process that returns the complexed form DlC to the free forms $cDl^0$ and $C_f$, making the DlCT gradient strongly correlated to the Toll activation profile.

Carrell *et al.*, 2017 [11] have shown that photoactivatable Dorsal spreads 6–7 nuclear diameters over 90 minutes when activated ventrally, but spreads to fill the entire view of the embryo in 90 min when activated on the dorsal side. Hence, the diffusion coefficient for Dl would range from $0.1\mu m^2/s$ on the ventral side to $5\mu m^2/s$ on the dorsal side. Therefore, the range of DlC diffusion from our model is within the range established by these authors. Also, Kanodia and collaborators [8] assumed that the DlC diffusion coefficient, free Dorsal and Cact are identical, resulting in a lateral transport coefficient of approximately one. This indicates that each of the molecular species above traverses a single compartment during cycle 14. Carrell *et al.*, 2017 [11], taking into account the changing distances between nucleo-cytoplasmic compartments due to mitosis, also find a Dl transport coefficient centered around 1. We have found a similar result for the lateral transport coefficient for DlC complex, since it travels 1–2 compartments over 90 minutes, according to the diffusion coefficients calculated above.

Despite the agreement between our results and the literature described above, the model suggests that DlC complex diffusivity is lower than free Dl and Cact (Table 1). In fact, the estimated diffusivity of Dl, Cact, $cDl^0$ and $C_f$ may be overestimated, since It has been previously reported that Dl diffusion among adjacent compartments takes place much more slowly than diffusion within individual nucleo-cytoplasmic compartments, which should dominate over intercompartmental diffusion [49,18]. We believe that these high diffusivities reported by our model are maybe related to not taking into account the already reported high Cactus turnover [6,19]. Without this, the model needs to report these high diffusivities in order to assure a uniform distribution for $cDl^0$, $nDl^0$, Cact and DlC along the embryonic DV axis (S1A–S1D Fig), despite interacting with non-uniformly distributed DlCT and $nDl^*$. Also, other aspects reported in other models and not assumed here are the Toll receptor saturation and shuttling. However, these particular aspects of our models do not impact the main focus of our analysis and the result is the signaling responsible for the role of Cact in the Toll-Dorsal signaling mechanism.

## A conserved regulatory network for Cactus/IκB function

Toll was initially described in *Drosophila* embryos as a maternal effect allele regulating DV patterning [50]. However, it has been suggested that the function of Toll to pattern the DV axis is restricted to the insect lineage while regulating the innate immune response is an ancient and widespread function [51,52]. In either context, elements of the Toll pathway are conserved in Bilateria. Conservation is not restricted to elements downstream of the Toll receptor. Toll-dependent and -independent pathways have been reported to regulate vertebrate IκB, as shown for *Drosophila* Cact. In mammalian cells, it has been proposed that constitutive degradation of IκBα independent of the proteasome regulates basal levels of IκB and thus the duration of the NFκB response [53–56]. Experimental data and modeling of signal independent IκB regulation suggest that the regulatory pathway controlling free IκB is a major determinant of constitutive NFκB and stimulus responsiveness of the NFκB signaling module [57]. Furthermore, calcium-dependent Calpain proteases also target mammalian IκB independent of Toll receptors [55,56,58], as reported for Cact in the embryo and immune system [17,26]. Unlike *Drosophila*, vertebrates rely on several IκB proteins, where the final effect on the immune response over time is a result of the compounded effect of three or more IκBs [59]. Interestingly, it has been shown that IκBβ knockout mice display a dramatic reduction of TNFα in response to lipopolysaccharide (LPS), suggesting that IκBβ acts to both inhibit and activate gene expression [60]. Even though the molecular mechanism proposed by Rao *et al.*, 2010, for

I$\kappa$B$\beta$ is different from the here presented for Cact, they are similar in the sense that both are able to improve NF$\kappa$B family transcriptional activity. Therefore, there is an open avenue of investigation on the mechanisms that regulate Cact/I$\kappa$B function with potentially important outcomes for immunity. Altogether, the results herein presented credit to I$\kappa$B/Cact protein a key role in NF$\kappa$B activity greater than previously reported.

## Materials and methods

### Embryo sections and Dl gradient analysis

Transverse optical sections were obtained at 90$\mu$m from the anterior region of the cc14 embryos as in [29]. For nDl gradient visualization, mutant and control Histone-GFP embryos were mixed, fixed and processed concomitantly as in [17]. Primary antisera used were mono-clonal anti-Dl (7A4; 1:100; DSHB) and anti-GFP (1:1000; Novus Biologicals, to detect control gradients). Dl target genes were visualized by *in situ* hybridization as in [26]. In all experiments, a control was included together with mutant embryos throughout the fixation, immunolabeling and imaging protocol, in order to minimize experimental variability that could potentially impact quantitative measurements. The mutant conditions analyzed were: embryos from $cact^{A2}/cact^{011}$, $cact^{A2}/dl^6$, and $dl^6/+$ mothers.

### Mathematical modeling of Dl nuclear gradient formation

The proposed reaction-diffusion model is a refinement of [8], in which the authors described Dl and Cact dynamics by a reaction-diffusion regulatory network along a 1D spatial domain that represents the embryo's outline in a dorsal-to-ventral cross-section. The pioneer model was used and modified by others [11,42]. Here we expanded this model to take into account the translational control of Cact levels by Dl protein and the two different routes for Dorsal nuclear localization.

We assumed that the embryo is symmetric with respect to the DV axis, therefore, all simulations acknowledge only one of the embryo's dorsal-ventral cross-section half, considering no-exchange boundary conditions at both ventral and dorsal midlines. The number of compartments is fixed and their sizes are equal, such that the length of the whole simulation region is one, for computational convenience.

Dorsal dimerization was not described in the model, since nuclear input dynamics are restricted to the dimeric form. Thus, although the model refers simply to Dl, it should be kept in mind that, for the purposes of this work, the species in question is always the corresponding dimer. The association of cytoplasmic Dorsal dimer (cDl$^\circ$) with Cactus (C$_f$) is also described and the resulting complex is treated as a new species (DlC). In addition, direct nuclear Dorsal import and export is treated as a reversible reaction in which nuclear Dorsal is treated as a new nuclear species (nDl$^\circ$).

The Toll receptor (T) in the model is the one activated by Spätzle, and hence its distribution along the dorsoventral region in the embryo is not uniform. In this work, it was assumed that its distribution is Gaussian, whose mean value is in the most ventral part of the embryo, and its standard deviation and intensity peak are two parameters fitted to experimental data, along with the kinetic constants and the diffusion coefficients.

In addition to participating in the reactions (Fig 1), some of the proteins can diffuse along the compartments. In particular, cDl$^\circ$, C$_f$ and DlC can diffuse following Fick's Law whereas cDl$^*$ and C$_{ub}$ reaction dynamics are considered much faster than their diffusion capacities, hence they do not have diffusion coefficients associated with them. All of the processes, which are modeled as chemical reactions, take place inside each compartment and are described by

the Mass Action Law. Further details about the resulting differential equations are displayed in S1 Text.

## Model calibration and simulation

To determine the model parameters, we calibrated kinetic parameters k1 to k12, $cDl^{\circ}$, $C_f$ and DlC diffusion coefficients, total Dl concentration, and Toll receptor intensity peak and standard deviation (assuming that its activation follows a Gaussian function centered at the most ventral region of the embryo) using experimental data of nuclear mitotic cycle 14 wild-type *Drosophila* embryos [29]. We set up an optimization problem such that the objective function is the quadratic loss where the model total nuclear Dl (the sum of $nDl^0$ and $nDl^*$) was compared to experimental nuclear Dorsal data. The sum of the quadratic differences between simulation and experimental data was minimized. The experimental data used was normalized so that the peak of nDl concentration at the most ventral midline is 1.

Since the loss function is not explicitly defined in terms of the parameter because it results from the numerical solution of a series of Ordinary Differential Equations (ODEs) with mass transfer terms between compartments, metaheuristics become convenient for intelligently exploring parameter possibilities, saving computational cost to obtain a viable set of parameters. Among several techniques, we have adopted the Genetic Algorithm (GA), which was implemented using a binary codification of the parameters, a selection by a k-tournament, a uniform bit-wise crossover and a mutation by bit-wise inversion [61,62].

For each individual in the Genetic Algorithm population, which corresponds to a set of parameters to be evaluated and ranked by the objective function, we have described the series of ODEs with mass transfer terms by taking the PDEs reported in S1 Text, applying a second-order central approximation to discretize spatial coordinates, giving rise to an ODE system for each compartment. Subsequently, a linear implicit multistep method based on 6th order finite differences (BDF-6 formula) was used to solve each ODE in time [63,64]. The computation considers, in the beginning, the species $cDl^0$ with uniform concentration given by the parameter [Dl]tot for each compartment, the species T following a Gaussian curve, as described in the S1 Text, and all other species with zero concentration, and it ended when the system had reached a steady-state for each individual in the GA population.

After running the program and finding out an optimized set of parameters, the mutant simulations have been done by changing specific parameters and comparing the newer simulations with a determined set of experimental data [29]. For *dl6/+* mutants, we have only changed total Dl concentration; for *cact*-related mutants, kinetic constant k1, which is related to Dl regulation of Cact, was changed; and both of these parameters were changed for $dl^6/cact^{A2}$ mutant. Further details about the equations, GA's parameters and other specifications can be found in S1 Text.

Due to the stochastic behavior of the Genetic Algorithm, running the model calibration several times generated different solutions. To assess the stability of the parameter set obtained, we constructed histograms for each parameter from a population whose cost function was less than or equal to 0.069 (as a reference, the best individual chosen for the model simulation shows a cost function of 0.015—see S6 Fig).

As expected, the parameters did not show a Gaussian distribution since the parameters are not independent of each other. The kinetic constants are correlated nonlinearly through the chemical reactions that define them. The histograms show that most of the best individuals found by the Genetic Algorithm show a narrow distribution in the parameter space, indicating considerable stability of the solution. This is even more significant considering that we used an epidemic strategy to prevent the population from being dominated by solutions within a local minimum.

It is noteworthy that the kinetic parameters, in general, have a narrow distribution, although some, such as k1 and k12, exhibit an even narrower distribution. This is also verified for the diffusion coefficients, total Dl concentration, peak intensity and width of activated Toll curve. The critical role of the Toll activation curve shape is reflected in the narrow distribution of the activated Toll width since its value ranges from approximately 0.15 to 0.20. Keeping in mind that the Genetic Algorithm does not guarantee that the solution obtained is the parameters global minimum, our goal is to verify the statistical representativity of the best individual chosen. This is critical to support the model capacity of reproducing the experimental data and its predictive character. More detailed studies of the probability distribution of the parameters can be done in the future based on the results of the Genetic Algorithm as a starting point.

## Typical time scale of the model and diffusion coefficients

In order to make the parameters found by the model comparable to the processes in the biological system with experimentally known time scales, it is necessary to recover the dimensions of the constants and coefficients obtained by our fitting procedure. The typical spatial scale of the system is the embryo half-length (L = 245μm, data obtained in Ambrosi *et al.*, 2014 [48]). However, the typical time scale of our model is more challenging to be found, since the fitting is concerned only about steady-state data, with artificial initial conditions and therefore cannot provide a realistic insight of the time scale until reaching the final state of the system.

Thus, one way to carry out such a calculation is to choose a physical process whose time scale is known or estimated, and determine which time scale is necessary for our model to suit the selected process. In particular, we chose the Dl nuclear import, whose time scale reported in Carrell *et al.*, 2017 is in the range of 2 min to 5 min [11]. Thus, the effective kinetic constant for Dorsal entry into the nucleus must be between $1/5$ min$^{-1}$ ($1/300$ s$^{-1}$) and $1/2$ min$^{-1}$ ($1/120$ s$^{-1}$).

As Dorsal enter in the nucleus by two different routes (represented by the constants k3 and k11), the effective constant can be determined by a concentration-weighted kinetic constants average (k3 modulated by cDl0, and k11 modulated by cDl*, considering the results at the most ventral region), hence obtaining the value of 0.0373. Thus, to achieve the effective constant discussed above, the time scale of our model must be between 4.5s and 11.19s.

With this, we can determine the diffusion coefficients in um$^2$ / s, multiplying the values of $D_{Dl}$, $D_C$ and $D_{DlC}$ by $(245μm)^2$ and dividing by the values obtained from the time scale. Note that, in this way, we obtain the following values, respectively: $4.9 \times 10^5$ μm$^2$/s—$1.2 \times 10^6$ μm$^2$/s for cDl0, $2.2 \times 10^6$ μm$^2$—$5.5 \times 10^6$ μm/s$^2$ for Cf, and $0.18$ μm$^2$/s—$0.45$ μm$^2$/s for DlC.

## Supporting information

**S1 Fig. Model prediction of the dorso-ventral distribution of all model species for wild type and *dl⁶*/+ mutant simulation.** Distribution of free cytoplasmic (cDl$^0$, A) and nuclear (nDl$^0$, B) Dorsal, free Cactus (C$_f$, C), DlC complexes formed by Dl dimer and Cact monomer (D), Toll (T) receptor (E), DlCT complexes including DlC and a Toll receptor (F), Cactus (C$_{ub}$, G) and cytoplasmic Dorsal (cDl*, H) modified by Toll Pathway (G-H), nDl modified by Toll dependent pathway (I).
(TIF)

**S2 Fig. Model prediction of the dorso-ventral distribution of all model species for wild type and *cact$^{A2}$*/*cact$^{011}$* mutant.** Distribution of free cytoplasmic (cDl$^0$, A) and nuclear (nDl$^0$, B) Dorsal, free Cactus (C$_f$, C), DlC complexes formed by Dl dimer and Cact monomer (D), Activated Toll (T) receptor (E), DlCT complexes including DlC and an activated Toll receptor

(F), Cactus ($C_{ub}$, G) and cytoplasmic Dorsal (cDl*, H) modified by Toll Pathway (G-H), nDl modified by Toll induced (I).
(TIF)

**S3 Fig. Model prediction of the dorso-ventral distribution of all model species for wild type and *dl⁶/cact^{A2}* mutant.** Distribution of free cytoplasmic (cDl⁰, A) and nuclear (nDl⁰, B) Dorsal, free Cactus ($C_f$, C), DlC complexes formed by Dl dimer and Cact monomer (D), activated Toll (T) receptor (E), DlCT complexes including DlC and an activated Toll receptor (F), Cactus ($C_{ub}$, G) and cytoplasmic Dorsal (cDl*, H) modified by Toll Pathway (G-H), nDl modified by Toll induced (I).
(TIF)

**S4 Fig. Model prediction of the dorsal-ventral distribution of all model species.** Curves for wild-type (black), *dl⁶/+* (orange), *cact^{A2}/cact^{011}* (blue), and *dl⁶/cact^{A2}* (purple) mutants. Distribution of free cytoplasmic (cDl⁰, A) and nuclear (nDl⁰, B) Dorsal, free Cactus ($C_f$, C), DlC complexes formed by Dl dimer and Cact monomer (D), activated Toll (T) receptor (E), DlCT complexes including DlC and an activated Toll receptor (F), Cactus ($C_{ub}$, G) and cytoplasmic Dorsal (cDl*, H) modified by Toll Pathway (G-H), nDl modified by Toll induced (I).
(TIF)

**S5 Fig. Model simulations recover the dual effect of Cactus on Dorsal in *cact^{A2}/cact^{011}* mutant background.** (A-C) Model prediction of nuclear species distribution for 3x, 2x or 1.5x increases of kinetic constant k7 in a *cact^{A2}/cact^{011}* mutant background. nDl, total nuclear Dorsal dimers (A); nDl⁰, free nuclear Dorsal dimers (B); nDl*, nuclear Dorsal dimers induced by activated Toll (C). (D) nDl simulation decreasing k2 by 20%, 30%, 40% or nulling k2 (0%) in a *cact^{A2}/cact^{011}* mutant background comparing to experimental data (circle symbol) and WT and *cact^{A2}/cact^{011}* mutant simulations. Simultaneous reduction of k2 and k5 by 0%, 10% and 50% (E) or by 0.1%, 0.01%, 0.001% and 0.0001% (H) comparing to control (WT) nDl concentration simulations (E, H). Reduction of k2 (F) or k5 (G) by 0%, 10% and 50%.
(TIF)

**S6 Fig. Histogram showing the distribution of different model parameters over selected individuals in the Genetic Algorithm.** We selected the 37 best individuals from the full Genetic Algorithm execution, following the criterion of the cost function being less than or equal to 0.069. (A-C) Diffusion coefficients, in logarithmic scale; (D-O) kinetic constants, in logarithmic scale; (P) peak activated Toll concentration; (Q) activated Toll width; (R) total Dorsal concentration.
(TIF)

**S7 Fig. Comparison between the normalized nuclear Dl gradient of wild-type and dl6/+.** Model concentration distribution of nuclear Dl (dotted lines) was normalized in order to compare the shape of the profiles obtained for the wild-type and dl6/+ genotypes. As a reference, Gaussian curves were fitted to the experimental data (solid lines).
(TIF)

**S1 Text. A detailed description of the model equations, loss function, and the genetic algorithm used in parameter calibration.**
(PDF)

**S1 Table. Quantitative data for nuclear Dorsal, extracted from cycle 14 Drosophila embryos that were fixed and processed for immunohistochemistry with anti-Doral and anti-GFP antibodies as in [17, 29].** All embryos were processed concomitantly and arranged

in a microfluidics device [65] for transversal slice image capture in a confocal microscope. After image capture, Nuclear fluorescence levels were extracted based on a Matlab routine as in [17,29,66], with the nuclei identified by DAPI staining, and arranged along the DV axis to generate half Gaussian curves. In the following sheets, nuclei are depicted according to their position along the dorsal (D) ventral (V) axis.

(XLSX)

## Acknowledgments

We are grateful to Trudi Schupbach for her helpful comments on the manuscript and to the reviewers for excellent suggestions that improved our manuscript. PB and HA would also like to thank excellence in research fellowships from CNPq—Conselho Nacional de Desenvolvimento Científico e Tecnológico (https://www.gov.br/cnpq/pt-br).

## Author Contributions

**Conceptualization:** Maira A. Cardoso, Paulo M. Bisch, Helena M. Araujo, Francisco J. P. Lopes.

**Data curation:** Claudio D. T. Barros, Maira A. Cardoso, Paulo M. Bisch, Helena M. Araujo, Francisco J. P. Lopes.

**Formal analysis:** Claudio D. T. Barros, Maira A. Cardoso, Paulo M. Bisch, Helena M. Araujo, Francisco J. P. Lopes.

**Funding acquisition:** Paulo M. Bisch, Helena M. Araujo, Francisco J. P. Lopes.

**Investigation:** Claudio D. T. Barros, Maira A. Cardoso, Paulo M. Bisch, Helena M. Araujo, Francisco J. P. Lopes.

**Methodology:** Claudio D. T. Barros, Maira A. Cardoso, Paulo M. Bisch, Helena M. Araujo, Francisco J. P. Lopes.

**Project administration:** Paulo M. Bisch, Helena M. Araujo, Francisco J. P. Lopes.

**Resources:** Paulo M. Bisch, Helena M. Araujo, Francisco J. P. Lopes.

**Software:** Claudio D. T. Barros, Francisco J. P. Lopes.

**Supervision:** Paulo M. Bisch, Helena M. Araujo, Francisco J. P. Lopes.

**Validation:** Claudio D. T. Barros, Maira A. Cardoso, Paulo M. Bisch, Helena M. Araujo, Francisco J. P. Lopes.

**Visualization:** Maira A. Cardoso, Helena M. Araujo.

**Writing – original draft:** Claudio D. T. Barros, Maira A. Cardoso, Paulo M. Bisch, Helena M. Araujo, Francisco J. P. Lopes.

**Writing – review & editing:** Claudio D. T. Barros, Maira A. Cardoso, Paulo M. Bisch, Helena M. Araujo, Francisco J. P. Lopes.

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
