## [Decision Letter · Decision Letter 0]

24 Mar 2020

Dear Dr Lopes,

Thank you very much for submitting your manuscript "A Reaction-Diffusion Network model predicts a dual role of Cactus/IκB to regulate Dorsal/NFκB nuclear translocation in Drosophila" (PCOMPBIOL-D-20-00145) for consideration at PLOS Computational Biology. As with all papers peer reviewed by the journal, your manuscript was reviewed by members of the editorial board and by several independent peer reviewers. Based on the reports, we regret to inform you that we will not be pursuing this manuscript for publication at PLOS Computational Biology.

The reviews are attached below this email, and we hope you will find them helpful if you decide to revise the manuscript for submission elsewhere. We are sorry that we cannot be more positive on this occasion. We very much appreciate your wish to present your work in one of PLOS's Open Access publications. 

Thank you for your support, and we hope that you will consider PLOS Computational Biology for other submissions in the future.

Sincerely,

David Umulis

Associate Editor

PLOS Computational Biology

Jason Haugh

Deputy Editor

PLOS Computational Biology

Reviewer's Responses to Questions

**Comments to the Authors: **

Reviewer #1: This paper refines previous mathematical models of Dorsal/NFkB gradient formation in Drosophila. The authors claim that their new model predicts a dual role of Cactus/IkB in inhibiting and promoting Dorsal nuclear translocation. The model is inspired by experimental observation with certain hypomorphic cactus allele combinations (Cardoso et al., 2017). New mechanisms included in the proposed model are the Dorsal-dependent translational control of Cactus (I could not trace the experimental evidence for this), the processing of Cactus (probably CalpA-dependent) and a Toll-dependent mechanism by which Cactus promotes Dorsal nuclear transport. The model has 18 parameters (12 kinetic constants, 3 diffusion coefficients, the total Dl concentrations and 2 activated-Toll related parameters). These are fitted to reproduce the experimentally measured Dorsal nuclear distribution using a genetic algorithm. Does this procedure always lead to the same stable outcome? How much parameter variation did the authors observe upon repeated runs of the optimization process?

My main criticism however concerns the fact that the authors did not correctly incorporate published literature. Their crucial simulation of the cactus null condition (Figure 4 E) leads to a uniform nuclear Dorsal distribution (corresponding to dorsolateral DV positional values!). However, this cannot be the case. They cite the paper by Bergmann et al., 1996 in which independent germ line clone experiments with three amorphic loss of function alleles of cactus were described. One of them is a defined genomic deletion of the entire cactus gene. In all three cases Bergmann et al observed nuclear Dorsal gradients, albeit with a decreased slope. Thus, polarity was not lost. Most importantly there was no indication of decreased ventral nuclear Dorsal concentrations compared to wildtype. Early germline clones by pole cell transplantations using one of these amorphic alleles (a cactus point mutation with an early stop codon deleting all ankyrin repeats) led to embryos with slightly expanded twist domains (Roth et al. 1991). Further fundamental mistakes of the current paper are the comparative simulations of pelle and Toll KD embryos. The experimentally observed null mutations of these genes lead to identical phenotypes with embryos lacking nuclear Dorsal to the same degree (Roth et al., 1989). It seems that the authors assume that Tollrm9 is a null allele, which in fact it is lateralized Toll hypomorph. In their simulation the null mutants of Cactus and that of Toll lead to almost identical phenotypes which both do not agree with experimental observations. The fact that their simulations lead to results agreeing with the wrong phenotypes, casts serious doubt on the entire approach. 

All available evidence suggests that Dorsal protein (monomer or dimer) can be subject to Toll induced phosphorylation in absence of Cactus and that this process is required to achieve the highest nuclear Dorsal concentrations. This explains why cactA2; Toll double mutants are lateralized at a ventrolateral level, while cactA2; Tollrm9 is mesodermalized (uniform highest nuclear Dorsal) (Roth et al., 1989).

Reviewer #2: Barros et al used a modeling approach to characterize the kinetics of nuclear Dorsal (Dl) gradient in early Drosophila embryos, and showed that the Dorsal-Cactus complex formation independent of Toll-signaling pathway is a key step to be added in the model to fully reproduce experimental data on nuclear Dl expression. Typically, it is known that Cactus (Cact) binds to Dl and blocks Dl’s entry into the nucleus. Upon activation of Toll signaling pathway, Cact is phosphorylated and degraded, allowing freed Dl to enter the nucleus. The authors had recently shown that Cactus actually has an additional role to promote Dl nuclear translocation, especially in the ventral region where Toll signal is high. Since the mechanism of Cact’s dual role in nuclear Dl regulation is not known, they used a new reaction-diffusion based model to characterize Dl dynamics. Using the model, they were able to show that in addition to Toll-dependent Dl translocation, a fraction of Dl-Cact complexes work independently of Toll pathway, and unbinding of Cact from the complex would allow direct entry of Dl into nucleus. This Toll-independent Cact regulation was crucial to elucidate the dual role of Cact – prevent and promote translocation of Dl protein. In cact mutant embryos, reduced Cact level would form fewer Dl-Cact complex, and consequently reduced number of Dl molecules can be freed from the Dl-Cact complex by activated Toll pathway, resulting in lower Dl level in cact mutant embryos, especially in ventral region. In the lateral/dorsal region where Toll signal is low, however, reduced Cact would allow more free Dl to directly enter nucleus, increasing basal Dl level in cact mutant embryos (dominant Toll-independent Cact regulation). Their model agrees with experimental data, including WT and various dl and cact mutant conditions. The manuscript is written logically, and dl/+, cact/cact, and dl/cact mutant data were used appropriately to test dual role of Cact and to better understand underlying mechanism of Cact regulation. 

Yet, there are some comments and questions I have on the manuscript that could improve this manuscript’s quality and clarity.

1) In Fig 7C, and SF1, how significant are these changes in basal levels of nDl or Cactus? For example, is the difference between 0.002 vs 0.003 significant (the authors claim that in some figure panels)? Does the trend hold with respect to some changes in parameters, or is this very sensitive to a certain set of parameters only?

2) Regarding the first question, how robust is the parameters with respect to changes?

3) Based on which data did authors use lower Cactus concentration to simulate dl6/+ condition in the model? I understand lower Dl level, but I missed the evidence for lower Cactus level in dl mutant.

4) In Fig 3D, why does nDl level lower in lateral/dorsal side as well (it wasn’t so in dl6/+)? Is there any experimental data to support the model data? 

5) In the author’s previous paper (2017 Development), the authors observed lateral expansion of Dl in cact mutant embryos. But, I don’t see such evidence in the new experimental data in Fig 4B. Any comment on that?

6) It took a while to understand how DlCT and DlC explains dual role of Cact. It would be nice if the authors clarify it a bit more. Personally, I understood it only after reading Discussion section. Maybe use some of these phrases in Results, when they first talked about it?

**Have all data underlying the figures and results presented in the manuscript been provided?**

Reviewer #1: Yes

Reviewer #2: Yes

PLOS authors have the option to publish the peer review history of their article (what does this mean?). If published, this will include your full peer review and any attached files.

Reviewer #1: No

Reviewer #2: No

---

## [Decision Letter · Decision Letter 1]

22 Sep 2020

Dear Dr Lopes,

Thank you very much for submitting your manuscript "A Reaction-Diffusion Network model predicts a dual role of Cactus/IκB to regulate Dorsal/NFκB nuclear translocation in Drosophila" for consideration at PLOS Computational Biology.

As with all papers reviewed by the journal, your manuscript was reviewed by members of the editorial board and by several independent reviewers. In light of the reviews (below this email), we would like to invite the resubmission of a significantly-revised version that takes into account the reviewers' comments.

Please note that this manuscript went out to additional reviewers. The first two reviews are copied into this as well. Note that for this to be considered further, all points raised by the reviewers will need to be addressed.

We cannot make any decision about publication until we have seen the revised manuscript and your response to the reviewers' comments. Your revised manuscript is also likely to be sent to reviewers for further evaluation.

Sincerely,

David Umulis

Associate Editor

PLOS Computational Biology

Jason Haugh

Deputy Editor

PLOS Computational Biology

Reviewer #1: This paper refines previous mathematical models of Dorsal/NFkB gradient formation in Drosophila. The authors claim that their new model predicts a dual role of Cactus/IkB in inhibiting and promoting Dorsal nuclear translocation. The model is inspired by experimental observation with certain hypomorphic cactus allele combinations (Cardoso et al., 2017). New mechanisms included in the proposed model are the Dorsal-dependent translational control of Cactus (I could not trace the experimental evidence for this), the processing of Cactus (probably CalpA-dependent) and a Toll-dependent mechanism by which Cactus promotes Dorsal nuclear transport. The model has 18 parameters (12 kinetic constants, 3 diffusion coefficients, the total Dl concentrations and 2 activated-Toll related parameters). These are fitted to reproduce the experimentally measured Dorsal nuclear distribution using a genetic algorithm. Does this procedure always lead to the same stable outcome? How much parameter variation did the authors observe upon repeated runs of the optimization process?

My main criticism however concerns the fact that the authors did not correctly incorporate published literature. Their crucial simulation of the cactus null condition (Figure 4 E) leads to a uniform nuclear Dorsal distribution (corresponding to dorsolateral DV positional values!). However, this cannot be the case. They cite the paper by Bergmann et al., 1996 in which independent germ line clone experiments with three amorphic loss of function alleles of cactus were described. One of them is a defined genomic deletion of the entire cactus gene. In all three cases Bergmann et al observed nuclear Dorsal gradients, albeit with a decreased slope. Thus, polarity was not lost. Most importantly there was no indication of decreased ventral nuclear Dorsal concentrations compared to wildtype. Early germline clones by pole cell transplantations using one of these amorphic alleles (a cactus point mutation with an early stop codon deleting all ankyrin repeats) led to embryos with slightly expanded twist domains (Roth et al. 1991). Further fundamental mistakes of the current paper are the comparative simulations of pelle and Toll KD embryos. The experimentally observed null mutations of these genes lead to identical phenotypes with embryos lacking nuclear Dorsal to the same degree (Roth et al., 1989). It seems that the authors assume that Tollrm9 is a null allele, which in fact it is lateralized Toll hypomorph. In their simulation the null mutants of Cactus and that of Toll lead to almost identical phenotypes which both do not agree with experimental observations. The fact that their simulations lead to results agreeing with the wrong phenotypes, casts serious doubt on the entire approach.

All available evidence suggests that Dorsal protein (monomer or dimer) can be subject to Toll induced phosphorylation in absence of Cactus and that this process is required to achieve the highest nuclear Dorsal concentrations. This explains why cactA2; Toll double mutants are lateralized at a ventrolateral level, while cactA2; Tollrm9 is mesodermalized (uniform highest nuclear Dorsal) (Roth et al., 1989).

Reviewer #2: Barros et al used a modeling approach to characterize the kinetics of nuclear Dorsal (Dl) gradient in early Drosophila embryos, and showed that the Dorsal-Cactus complex formation independent of Toll-signaling pathway is a key step to be added in the model to fully reproduce experimental data on nuclear Dl expression. Typically, it is known that Cactus (Cact) binds to Dl and blocks Dl’s entry into the nucleus. Upon activation of Toll signaling pathway, Cact is phosphorylated and degraded, allowing freed Dl to enter the nucleus. The authors had recently shown that Cactus actually has an additional role to promote Dl nuclear translocation, especially in the ventral region where Toll signal is high. Since the mechanism of Cact’s dual role in nuclear Dl regulation is not known, they used a new reaction-diffused based model to characterize Dl dynamics. Using the model, they were able to show that in addition to Toll-dependent Dl translocation, a fraction of Dl-Cact complexes work independently of Toll pathway, and unbinding of Cact from the complex would allow direct entry of Dl into nucleus. This Toll-independent Cact regulation was crucial to elucidate the dual role of Cact – prevent and promote translocation of Dl protein. In cact mutant embryos, reduced Cact level would form fewer Dl-Cact complex, and consequently reduced number of Dl molecules can be freed from the Dl-Cact complex by activated Toll pathway, resulting in lower Dl level in cact mutant embryos, especially in ventral region. In the lateral/dorsal region where Toll signal is low, however, reduced Cact would allow more free Dl to directly enter nucleus, increasing basal Dl level in cact mutant embryos (dominant Toll-independent Cact regulation). Their model agrees with experimental data, including WT and various dl and cact mutant conditions. The manuscript is written logically, and dl/+, cact/cact, and dl/cact mutant data were used appropriately to test dual role of Cact and to better understand underlying mechanism of Cact regulation.

Yet, there are some comments and questions I have on the manuscript that could improve this manuscript’s quality and clarity. I think the manuscript is appropriate to be published in PLOS Computational Biology, with minor revision.

1) In Fig 7C, and SF1, how significant are these changes in basal levels of nDl or Cactus? For example, is the difference between 0.002 vs 0.003 significant (the authors claim that in some figure panels)? Does the trend hold with respect to some changes in parameters, or is this very sensitive to a certain set of parameters only?

2) Regarding the first question, how robust is the parameters with respect to changes?

3) Based on which data did authors use lower Cactus concentration to simulate dl6/+ condition in the model? I understand lower Dl level, but I missed the evidence for lower Cactus level in dl mutant.

4) In Fig 3D, why does nDl level lower in lateral/dorsal side as well (it wasn’t so in dl6/+)? Is there any experimental data to support the model data?

5) In the author’s previous paper (2017 Development), the authors observed lateral expansion of Dl in cact mutant embryos. But, I don’t see such evidence in the new experimental data in Fig 4B. Any comment on that?

6) It took a while to understand how DlCT and DlC explains dual role of Cact. It would be nice if the authors clarify it a bit more. Personally, I understood it only after reading Discussion section. Maybe use some of these phrases in Results, when they first talked about it?

Reviewer #3: The model by Barros et al. focuses on determining what the roles might be of novel aspects of Dl/Cact interactions. Namely, they focused on the Toll-independent regulation of Cact, the Dl-dependent translation of Cact, and the differentiation between “activated” Dl and “non-activated” Dl. The Araujo lab has been discovering some of these novel behaviors of Cact for several years, and this paper appears particularly motivated by the discoveries in Cardoso et al., 2017, in which Cact was shown to have a dual role for the Dl gradient.

It must be absolutely stressed that this model is an important contribution to the field, and I am looking forward to seeing it published. Since the pioneering modeling paper by Kanodia et al. (2009), the Dl/Cact modeling community has completely ignored the role of phosphorylation of Dl by Toll in the formation of the Dl gradient. The role that Dl plays in upregulating Cact protein levels has also been summarily ignored, even though, (1) on the face of it, it appears to be a very interesting regulatory mechanism, and (2) it was discovered a quarter century ago. Thus, this model is needed.

In addition, I must also stress that the authors’ approach is very important to the field of developmental biology: quantitative experiments combined with hypothesis-testing through mathematical modeling. While at times, new quantitative results may seem to be at odds with classical experimental observations (colorimetric reactions, cuticle preps, etc.), the quantitative nature of the work gives important nuance to phenotypes that were not previously appreciated, and at times, may serve as a corrective for incorrectly-interpreted qualitative observations.

All of that being said, the work has several drawbacks that still need to be addressed, both major and minor. Most of the major concerns stem from the authors ignoring or not taking into account all interactions that have been discussed in the Dl/Cact pathway. This is not generally a problem, as it is not strictly necessary for every new model to always have at least the same interactions as previously-published models. Along those lines, it can be argued that they did not intend the model to capture these interactions, and instead focused on the roles that their own novel contributions might play in shaping the Dl gradient. However, there are two issues with this: first, this work discusses aspects of the Dl gradient that are impacted by the interactions that were left out. And second, some of their conclusions, such as the values of diffusivity, are justified with faulty reasoning, and instead may simply stem from the authors not taking all data/published interactions into account. More details on all of this below, but the general ways to rectify this problem are (1) to directly discuss the pros and cons of their model and conclusions in regards to leaving out previously-published interactions, and (2) not to emphasize the conclusions that may stem from the fact that they left those interactions out. As such, most of my criticisms could be answered by adding crucial discussion elements and changing the emphasis of some parts of the text.

Before I proceed with detailed criticisms, I feel the need to state that as a reviewer, I am in a bit of an odd position, coming to the party during the second round of reviews. I will attempt to provide comments in light of both the first and second submissions of the paper, the first round of reviews, and the authors’ responses.

Major concerns (in no particular order):

1) The authors assume only “active” nuclear Dl can regulate target genes. The authors should state evidence for this. If no evidence is available, and this is an assumption by the authors, this would imply this “direct-flow” nDl stands in the place of n[DlC]: figures such as Fig 1D are very similar to Fig 3F from O’Connell and Reeves (2015). Both hypotheses account for the apparent discrepancy between the experimentally-measured levels of nuclear Dl and the need for the Dl *activity* gradient to drop to near zero levels. Thus, the authors should discuss the pros and cons of favoring their interpretation of “deconvolving” the nuclear Dl gradient into two components vs. the interpretation of Dl/Cact complex acting as the contributor to the basal levels of the gradient. The discussion should include how the phenotypes attributed to nuclear Dl/Cact complex could be explained by their approach or by alternative mechanisms.

2) One of the paper’s main results, that their model can account for the dual-role of Cact, has possibly been previously explained by the shuttling mechanism. The authors should discuss the pros and cons of favoring their interpretation of Cact’s dual role over shuttling, including how the phenotypes described by shuttling (Carrell et al., 2017 and Schloop et al., 2020) could be explained by alternative mechanisms.

One result of their simulations was that D_{DlC} is very low (more on this below). If true, this rules out shuttling. Therefore, if the authors wish to make that claim, it would be helpful to describe why having a low D_{DlC} is an important conclusion to their model. Alternatively, this result could be a side effect of the fact that they did not account for all known aspects of Dl/Cact interactions, or did not account for all experimental data. That would be acceptable, but if that is the case, the authors may want to acknowledge that and to downplay their result that D_{DlC} is low.

3) The authors investigate how reducing the maternal dl contribution affects the Dl gradient. Their endeavor has multiple drawbacks. First, their simulations fail to account for the change in shape of the Dl gradient: from roughly bell-shaped to flat-topped. As shown by Carrell et al., 2017, Toll saturation is a crucial mechanism that re-shapes the Dl gradient when Dl levels are reduced. Carrell et al. showed that without Toll saturation, all prior models predicted the Dl gradient to be exactly the same shape, regardless of the initial conditions on Dl. When normalized, wildtype Dl gradients lined up exactly on top of heterozygous ones. The authors should also attempt this normalization procedure to see if all of their reduced-[Dl] gradients (Fig. 3B,D) also fall on top of each other, and comment on the results.

Second, the authors should investigate how or why the dl[6]/+ simulations cannot match the experimental data. It is actually possible that Toll saturation is implicit in their model equations, which is all the more reason to try to adequately fit the dl[6]/+ data.

Third, the authors state that their simulations do in fact replicate the altered shape of the dl[6]/+ gradients (lines 206-209 of the tracked-changes document). This is clearly not the case, and this claim should be removed. Instead, the authors should acknowledge that their simulations do not match the observed shape change. The ability of the 1xdl Dl gradient to match the wildtype at 20% and 50% DV coordinate was explained in Al Asafen et al., 2020, and can be traced to a combination of Toll saturation, shuttling, and the presence of Dl/Cact complex in the nuclei.

4) In the authors’ words, the novel contributions of their model can be described as “two Dl nuclear translocation mechanisms, translational regulation of Cact by Dl, and the two pathways leading to Cact regulation: the Toll-regulated and Toll-independent pathways.” I could not find any analysis that attempted to determine which of these three processes is responsible for their most important results. On the face of it, I suspect that only the “two Dl nuclear translocation mechanisms” is important. In particular, the fact that only DlC, and not cDl0, can bind to Toll, is the main reason for their most important results. Thus, the dual-role for Cact can be traced back to Toll-dependent phosphorylation of Dl, which in their model is mediated by Cact.

Reviewer 1 also made mention of this, describing one of the novel aspects of the model as “a Toll-dependent mechanism by which Cactus promotes Dorsal nuclear transport.” The authors responded by saying, “We actually suggested the inverse, that a Toll Independent mechanism of Cactus regulation controls Dorsal nuclear import…In fact, we believe that we made clear in the manuscript that a Toll-independent mechanism is being described…” Unfortunately, the authors did not make that point clear. In fact, in the Discussion, the authors state, “In forming DlC, Cact plays a negative role to impair Dl from entering the nucleus, while by enabling new DlCT complexes it favors Dl nuclear translocation.” Perhaps I am misunderstanding what they are trying to say in the rebuttal to Reviewer 1.

As far as I can tell, the Toll-independent mechanism amounts to a first-order degradation of Cact, which is present in all Dl/Cact models up until this point. The novel regulation of Cact degradation comes from the fact that there is Toll-dependent production of C_ub, which is a species that has not previously been modeled. Indeed, previous models all assumed that Toll’s role in Cact regulation was to split apart the Dl/Cact complex, and free Cact was subsequently degraded by constitutive mechanisms (i.e., a high first order degradation rate), which are mechanistically indistinguishable from the authors’ Toll-independent pathway. Furthermore, the fact that only DlC can bind to Toll, which then promotes Dl nuclear translocation at a rate higher than “direct flow,” suggests that two aspects of the Toll-dependent mechanism are responsible for the promotion role for Cact: modeling of nDl* and the fact that only DlC can bind Toll.

In summary for this point, what is novel is *not* the Toll-independent pathway for Cact degradation. All models had this. What is novel is (1) the modeling of a Dl* species that has different properties from Dl0, (2) the fact that only DlC can bind Toll to create Dl*, and (3) the creation of a C_ub species. (Of these three, I am fairly certain the third point is not very important.) Therefore, I implore the authors first to analyze their model further to determine which of their novel aspects is important for their main results, and in particular which pathway (Toll-dependent or -independent) is responsible for the promoting role for Cact, so that their conclusions can be discussed without confusion. (This would be an important analysis to do even without the confusion.) And second, to devote less discussion to the Toll-independent pathway, as this is not the novel aspect (all previous models had this). Instead, the novel aspect is the modeling of Dl*, together with their description of Cact-mediated binding of DlC by Toll.

5) As mentioned above, the data-fitting results in a low DlC diffusivity. The authors state that this is “due to interaction with the Toll receptor,” implying that binding of DlC to Toll slows down its movement. This is not true. In their model, the diffusivity (intercompartmental exchange) of DlC is a free parameter that can be varied independently from DlC’s interactions with Toll. The *effective* or *apparent* diffusivity of DlC (which would be observed if watching the spread of DlC, and which might be characterized as D_{DlC}/(1 + Keq), where Keq is the equilibrium constant of DlC-Toll binding) could be low due to Toll binding, but not D_{DlC} itself. As a counter example, D_{Dl} is not low, even though Dl is captured by the nuclei; however, an *effective* diffusivity of Dl could be low. It seems strange they would explain the low value of D_{DlC} that way while at the same time acknowledging this does not apply to cDl, saying “the intercompartmental diffusion term…is much larger than the kinetic constants k3, k5 and k11.” At any rate, if D_{Dl} can be determined independently of the other kinetic parameters that might otherwise slow down Dl, so can D_{DlC}. The authors should revise their explanation for why D_{DlC} is low.

6) Along those lines, it is very surprising that the diffusivities of cDl and Cact dominate over other processes. First, intercompartmental exchange is slow (Delotto et al., 2007). Second, Cact is known to have a high turnover rate, so a slow intercompartmental exchange should not dominate a high degradation rate. Third, nuclear import of Dl, which has a time scale of roughly 1 min (eyeballing FRAP data from Delotto et al., 2007) or roughly 2-5 min (boxplots by Carrell et al., 2017), would be unlikely to be dominated by a process that apparently has a time scale of roughly 10 minutes. Indeed, the paGFP experiments by Carrell et al., 2017 suggest that nuclear capture dominates, as paGFP spreads only 6-7 nuclear diameters over 90 min on the ventral side, but spreads to fill the entire view of the embryo in 90 min when activated on the dorsal side (where no nuclear capture occurs).

7) Also about diffusivity: In dimensional units, either the diffusivity of Dl is itself too high (and not just in comparison to other processes), or the authors’ time unit is too large. They state that D_{Dl} = 91 inverse time units (D_{C} is even larger). To put this parameter back into dimensional units, one must multiply by the square of the half-circumference of the embryo (about 300 microns), then divide by T = 1, which is in an unspecified time unit. This gives a dimensional D_{Dl} of 8.91e6 um^2 per (time unit). While the authors do not specify what their time unit is, to bring a number of 9 million down to the expected order of 0.1-1 um^2/s, that would require their time unit to be about a year: 9 million um^2/yr roughly equals 0.3 um^2/s.

Of course, if the time unit were a year, nuclear import of Dl would be 1 inverse year (or 2e-6 inverse minutes). In which case, Dl would effectively never enter nuclei. (Thus, two criticisms are at play here: acknowledge the too-high value of D_{Dl} required by their model, and explicitly specify their time scale.)

Given the predictions of their model being at odds with “known wisdom” (as in the case of Cact turnover) and quantitative measurements (both from point #6), the too-high value of D_{Dl} (this point), as well as point #5, the authors need to revise their discussion subsection on diffusivity. In particular, the authors should acknowledge their model makes diffusivity predictions opposite of what would be expected from known data (as to which processes dominate), and discuss this in light of the fact that their model focus was only on their own important, novel contributions. They should also acknowledge that their model predicts a quantitatively too-high value of diffusivity. All of these are potentially fine, as long as the limitations are acknowledged and not presented as strengths or major predictions of the model. Not everything about every model needs to match with every expectation or previous experimental observation, especially if the model in question is leaving out previously-published interactions.

Minor concerns (in no particular order):

1) The authors present their model as a series of PDEs, which they then discretize to match the compartments. This is incorrect. Their model was never a series of PDEs, because they always intended to discretize space into 50 compartments to match the different cytoplasmic regions of the syncytial blastoderm. If the model really were PDE-based at its foundational level, then the model results should not depend on how space is discretized. They should be able to discretize space into 40 or 75 compartments and get the same results. Given the geometry of the embryo, the model is actually, at its foundational level, a series of ODEs, with mass transfer terms that resemble, but do not originate from, a discretized second derivative. This should be fixed.

2) The authors should report their values of D_{Dl}/L^2 and D_{Cact}/L^2 in scientific notation, potentially with periods meaning decimal points rather than separations among groups of three digits. The typical reader would wonder what “1.037.500” means. As it was, until I did the calculation (91*50^2), I thought their value of D_{Dl}/L^2 was two hundred twenty seven point five, instead of 2.27e5.

3) First sentence of the Results section, the authors claim that there are two different mechanisms for Dl nuclear transport. They further claim that cDl* is actively imported. I am not aware of a study that shows the *mechanisms* by which cDl0 and cDl* enter the nuclei are different, or that cDl* import is active. The paper they cite at the end of that sentence, Delotto et al., 2007, does not present any evidence that the mechanisms are different or that there is active transport. Indeed, in the very next sentence, the authors acknowledge there is no discrimination in mechanism in that paper, or any other paper for that matter. The authors should revise the first sentence to reflect this. Perhaps state that it has been shown that Dl* has a higher affinity for the nucleus (and cite the appropriate paper), which they take to mean the mechanisms of import are different, although there could be other interpretations (such as differing mechanisms of export).

4) Line 149 of revised submission with changes tracked: This is the only place that Fig. 2C is referenced in the manuscript. Fig. 2C has a depiction of gene expression domains, but nothing about gene expression is mentioned in the main text. The authors should elaborate.

5) Table 2: Why does the value of [Dl]tot change between dl[6]/+ and dl[6]/cact[A2]? Forgive me if this was answered elsewhere…I cannot find it.

6) Table 2, still: Can the authors elaborate on where the parameters [Dl]tot and [C]tot actually enter the model? Does [Dl]tot affect the initial conditions for Dl-containing species? What does [C]tot do? I could not find this information, so please forgive me if it was accessible.

7) Lines 288-290, tracked-changes file: The authors state that nDl0 and nDl* have different DNA binding efficiencies. Where are they getting this information? There is no citation. This is essentially the same question as my first “Major” criticism, but pointed out in a different place in the manuscript where a citation is needed.

8) Lines 163-164: What is the experimental nuclear Dl concentration in the ventral-most region? Do the authors have a number for that, and a citation?

9) The methods are lacking information about fly lines and experimental protocols. The extent of the Dl gradient in space appears to be closer to what was reported by Chung et al., 2011 and Kanodia et al., 2011, when the imaging was done only 70 microns from the pole. This has been clearly shown to result in a very extended Dl gradient. Is this how the embryo images in this paper were taken? What methods were used to fix and section the embryos? What antibodies were used? Also, the full genotypes of fly lines should be reported.

10) The Gaussian shape for the activated Toll gradient is in a bit of an unusual form. Typically, there is a “2” multiplying the sigma-squared term in the argument’s denominator. This may seem nitpicky, but leaving that out makes it more difficult to compare the width of their Toll domain to parameters used in other papers. An unobservant reader may not realize they are not equivalent parameters.

11) Fig. 6B: the authors characterize the Dl gradient’s ability to place boundaries by the slope of the gradient. In fact, the proper parameter is the slope divided by the concentration (1/c * dc/dx). Please revise accordingly.

**Have all data underlying the figures and results presented in the manuscript been provided?**

Reviewer #3: **No: **Data associated with images and the images that went into creation of graphs are not available. Nor is the code that went into their mathematical model.

PLOS authors have the option to publish the peer review history of their article (what does this mean?). If published, this will include your full peer review and any attached files.

Reviewer #3: **Yes: **Greg Reeves
---

## [Decision Letter · Decision Letter 2]

23 Mar 2021

Dear Dr Lopes,

Thank you very much for submitting your manuscript "A Reaction-Diffusion Network model predicts a dual role of Cactus/IκB to regulate Dorsal/NFκB nuclear translocation in Drosophila" for consideration at PLOS Computational Biology. As with all papers reviewed by the journal, your manuscript was reviewed by members of the editorial board and by several independent reviewers. The reviewers appreciated the attention to an important topic. Based on the reviews, we are likely to accept this manuscript for publication, providing that you modify the manuscript according to the review recommendations.

The reviewers have now gone through the revision and have remaining comments that were not adequately addressed in the previous rounds of review. Specifically, there are a number of instances with inappropriate claims of novelty that are pointed out by Reviewer 3. Furthermore, a more detailed expansion on areas where the model is unable to recapitulate experimental findings needs to be included. This specifically points to areas where the model does not predict expected patterns for cactus and Toll null mutations. This is only superficially addressed in the current manuscript, and hence more lucid explanations are needed to address these critical concerns.

Sincerely,

David Umulis

Associate Editor

PLOS Computational Biology

Jason Haugh

Deputy Editor

PLOS Computational Biology

[LINK]

The reviewers have now gone through the revision and have remaining comments that were not adequately addressed in the previous rounds of review. Specifically, there are a number of instances with inappropriate claims of novelty that are pointed out by Reviewer 3. Furthermore, a more detailed expansion on areas where the model is not able recapitulate experimental findings needs to be included. This specifically points to areas where the model does not predict expected patterns for cactus and Toll null mutations. This is only superficially addressed in the current manuscript and more lucid explanations are needed to address these critical concerns.

Reviewer's Responses to Questions

**Comments to the Authors:**

Reviewer #1: The authors produce a model for how Cactus is involved in Dorsal nuclear entry. Two routes are suggested. Route one requires activated Toll receptors and leads to phosphorylation and degradation of Cactus. Route two depends on direct reversible flow of free Dl dimers into the nucleus. The Dl dimers can associate with free Cactus, which itself is subject to Toll-independent processing. The authors first show that they can fine-tune the 18 parameters of their model so that their simulations reproduce the gradient of wt, dl heterozygosity, a hypomorphic cactus allele combination and cact/dl double heterozygotes. With model parameters fitting the data they try to simulate more a severe loss of cactus including the absence of cactus. This leads to a clear disagreement with published data and, according to the authors, suggests additional mechanisms not incorporated in the current model. However, this mechanisms uncovered by a null mutation are likely to contribute to the wt, dl heterozygous, cactus reduced and transheterozygous phenotypes they have simulated. Thus, the fact that their model fits to these data is a result of parameter fine-tuning and does not reflect the validity of the mechanistic assumptions.

The same applies to the authors attempt to simulate the loss of Toll activity. It makes perfect sense to take the reduction of DICT as a proxy for the loss of Toll activity. However, their simulation of the complete loss of Toll activity, i.e. DICT = 0 leads to lateralized phenotype, which again does not correspond to experimentally observed situation. I cannot see how the comparison to the special hypomorph Tlrm9 is helpful, here.

Taken together, the revision has not eliminated my main concerns about this study. The inability of the model to reproduce the phenotypes of cactus and Toll null alleles casts serious doubts about its basic assumptions.

Reviewer #3: In this submission of the manuscript, the authors addressed almost all of my previous criticisms. By adding more discussion, they presented the strengths of the model evenhandedly while still honestly acknowledging the small number of drawbacks of the model. However, there are still two criticisms that must be addressed. These criticisms are minor, in that they can be easily addressed by adding or deleting a few sentences, yet they are still important. Given they require such small edits, I trust the authors to be able to make the edits without needing to review the manuscript a third time.

I believe both of these criticisms stem from semantics, and after I present the details of the criticisms below, I offer a simple solution where the authors re-word a few things to clarify.

First and foremost, the authors continue to claim they have introduced a novel Toll-independent pathway for the degradation of Cactus. In my first review, I mentioned multiple times that all previous models had this. In their resubmission (the third, but only the second one that I reviewed), they added in multiple places (lines 30, 497, 500) the opposite statement: no other models had this. This is false: every other model had a term in the Cact equation that was the negative of a rate constant times the Cact concentration. Every other model assumed free Cact was degraded constitutively, and that Toll signaling simply split apart the Dorsal/Cactus complex to expose free Cact to the constitutive (Toll-independent) degradation.

The good news is that, as mentioned above, the authors introduce something else novel: the Toll-dependent pathway for the degradation of Cactus. No other model had Cact_ub as a species. So the novelty of their model, as far as Cact degradation is concerned, is the Toll-dependent ubiquitination of Cact, which allows for a different degradation rate than the Toll-independent degradation rate. So when the authors stated that no other model had the Toll-independent degradation of Cact, I think what they meant is that, since no other models had the Cact_ub species (which is the real novelty of the authors’ model), then no other model had the ability *to distinguish between* the two pathways.

Second, the same could be said for the semantics of free Cact binding to Dorsal. On line 503, they claim this is a novel term in their equations, but then admit it was previously found in other models while stating that previous models did not have it “as part of an independent pathway.” However, the term in the model that describes this interaction is the exact same as the corresponding term in all other models: a rate constant times the concentration of free Cact times the concentration of free Dorsal.

Again, the good news is that this model does have something different that no other model before it had: an “activated” form of Dorsal, which cannot be bound by Cact. So again, as the first criticism, the novelty is not in the Toll-independent pathway, but in the explicit modeling of the Toll-dependent effects (here modeling Dl*, and above modeling Cact_ub). Here this also allows the authors *to distinguish between* the two pathways.

One more point about this second criticism: the authors state, as part of this novel term, that they have modeled “Mobilization of free Cactus.” I cannot tell what they mean by “Mobilization;” it only appears one other place in the text (in a slightly different context). I suggest the authors remove this term.

In general, both of these criticisms seem to be semantic: the authors intended the Toll-independent mechanisms to be the novel parts of the model, but what they have in fact done is add novel components of Toll-dependent action. This is proven by the fact that the Toll-independent terms in their model show up in the exact same way in all other Dorsal/Cact models. However, they have two novel species in their model: Cact_ub and Dl*, which no other previous model had, and which are both dependent on Toll.

Perhaps a middle ground could be forged, in that the authors could state that their goal was to determine the relative contributions of the two pathways, and in order to do that, they had to distinguish them by explicitly modeling the Toll-dependent species. That would highlight what is novel about their model (Cact_ub and Dl*) while also highlighting what they think is important: the difference between the Toll-dependent pathway and the Toll-independent pathway. As such, I recommend the authors do the following:

- Instead of saying their Toll-independent terms are novel, say the Toll-dependent terms (species) are novel

- State the motivation for explicitly modeling the Toll-dependent species is that they wanted to determine the relative contributions of the Toll-independent vs Toll-dependent pathways, and that in order to do that, they needed the novel Toll-dependent species.

- Keep any language that states the importance of the Toll-independent pathway. Previous models could make no such claim, because they didn’t differentiate between the two pathways.

- To do this, they will have to edit lines 495 – 508 at the very least.

I should note here that, even though these criticisms stem from semantics, if the authors do not correct how they present the impact of their paper, many readers would be confused about the state of the field and how their paper fits into it. So the criticisms must be addressed.

One minor issue that is not associated with the above disagreements: the sentence starting on line 770 of the manuscript with changes highlighted should be edited. I believe the authors forgot to delete “Partial Differential Equations System (PDEs).”

Other minor typos could be found throughout, so the authors should edit for grammar.

**Have all data underlying the figures and results presented in the manuscript been provided?**

Reviewer #1: Yes

Reviewer #3: Yes

PLOS authors have the option to publish the peer review history of their article (what does this mean?). If published, this will include your full peer review and any attached files.

Reviewer #1: No

Reviewer #3: **Yes: **Greg Reeves

Figure Files:

Data Requirements:

Reproducibility:

References:

---

## [Editor Report · Decision Letter 3]

3 May 2021

Dear Dr Lopes,

We are pleased to inform you that your manuscript 'A Reaction-Diffusion Network model predicts a dual role of Cactus/IκB to regulate Dorsal/NFκB nuclear translocation in Drosophila' has been provisionally accepted for publication in PLOS Computational Biology.

Best regards,

David Umulis

Associate Editor

PLOS Computational Biology

Jason Haugh

Deputy Editor

PLOS Computational Biology

---

## [Editor Report · Acceptance letter]

21 May 2021

PCOMPBIOL-D-20-00145R3 

A Reaction-Diffusion Network model predicts a dual role of Cactus/IκB to regulate Dorsal/NFκB nuclear translocation in Drosophila

Dear Dr Lopes,

I am pleased to inform you that your manuscript has been formally accepted for publication in PLOS Computational Biology. Your manuscript is now with our production department and you will be notified of the publication date in due course.

With kind regards,

Agota Szep
